# Risk-averse Heteroscedastic Bayesian Optimization

**Anastasiia Makarova**
ETH Zürich
anmakaro@ethz.ch

**Ilnura Usmanova**
ETH Zürich
ilnurau@ethz.ch

**Ilija Bogunovic**
ETH Zürich
ilijab@ethz.ch

**Andreas Krause**
ETH Zürich
krausea@ethz.ch

## Abstract

Many black-box optimization tasks arising in high-stakes applications require risk-averse decisions. The standard Bayesian optimization (BO) paradigm, however, optimizes the expected value only. We generalize BO to *trade mean and input-dependent variance* of the objective, both of which we assume to be unknown a priori. In particular, we propose a novel risk-averse heteroscedastic Bayesian optimization algorithm (RAHBO) that aims to identify a solution with high return and low noise variance, while learning the noise distribution on the fly. To this end, we model both expectation and variance as (unknown) RKHS functions, and propose a novel risk-aware acquisition function. We bound the regret for our approach and provide a robust rule to report the final decision point for applications where only a single solution must be identified. We demonstrate the effectiveness of RAHBO on synthetic benchmark functions and hyperparameter tuning tasks.

## 1 Introduction

Black-box optimization tasks arise frequently in high-stakes applications such as drug and material discovery [17, 22, 28], genetics [16, 27], robotics [5, 12, 25], hyperparameter tuning of complex learning systems [13, 21, 34], to name a few. In many of these applications, there is often a trade-off between achieving high utility and minimizing risk. Moreover, uncertain and costly evaluations are an inherent part of black-box optimization tasks, and modern learning methods need to handle these aspects when balancing between the previous two objectives.

Bayesian optimization (BO) is a powerful framework for optimizing such costly black-box functions from noisy zeroth-order evaluations. Classical BO approaches are typically *risk-neutral* as they seek to optimize the expected function value only. In practice, however, two different solutions might attain similar expected function values, but one might produce significantly noisier realizations. This is of major importance when it comes to actual deployment of the found solutions. For example, when selecting hyperparameters of a machine learning algorithm, we might prefer configurations that lead to slightly higher test errors but at the same time lead to smaller variance.

In this paper, we generalize BO to trade off mean and input-dependent noise variance when sequentially querying points and outputting final solutions. We introduce a practical setting where *both* the black-box objective and input-dependent noise variance are *unknown* a priori, and the learner needs to estimate them on the fly. We propose a novel optimistic risk-averse algorithm – RAHBO – that makes sequential decisions by simultaneously balancing between *exploration* (learning about uncertain actions), *exploitation* (choosing actions that lead to high gains) and *risk* (avoiding unreliable actions). We bound the cumulative regret of RAHBO as well as the number of samples required to output a single near-optimal risk-averse solution. In our experiments, we demonstrate the risk-averse performance of our algorithm and show that standard BO methods can severely fail in applications where reliability of the reported solutions is of utmost importance.

35th Conference on Neural Information Processing Systems (NeurIPS 2021).

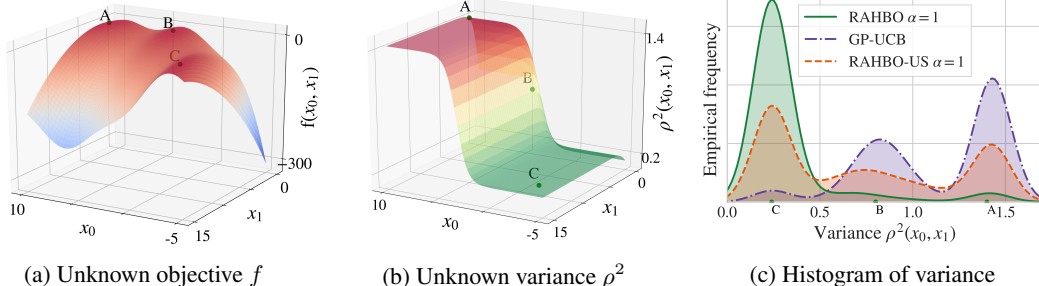

(a) Unknown objective $f$          (b) Unknown variance $\rho^2$          (c) Histogram of variance

Figure 1: When there is a choice between identical optima with different noise level, standard BO tends to query points corresponding to higher noise. (a) Unknown objective with 3 global maxima marked as (A, B, C); (b) Heteroscedastic noise variance over the same domain: the noise level at (A, B, C) varies according to the sigmoid function; (c) Empirical variance distribution at all points acquired during BO procedure (over 9 experiments with different seeds). The three bumps correspond to the three global optima with different noise variance. RAHBO dominates in choosing the risk-averse optimum, consequently yielding lower risk-averse regret in Figure 5a.

**Related work.** Bayesian optimization (BO) [26] refers to approaches for optimizing a noisy black-box objective that is typically expensive to evaluate. A great number of BO methods have been developed over the years, including a significant number of variants of popular algorithms such as GP-UCB [35], Expected Improvement [26], and Thompson Sampling [14]. While the focus of standard BO approaches is mainly on trading-off exploration vs. exploitation and optimizing for the expected performance, in this work, we additionally focus on the risk that is involved when working with noisy objectives, as illustrated in Figure 1.

The vast majority of previous BO works assume (sub-) Gaussian and *homoscedastic* noise (i.e., input independent and of some known fixed level). Both assumptions can be restrictive in practice. For example, as demonstrated in [15], the majority of hyperparameter tuning tasks exhibit heteroscedasticity. A few works relax the first assumption and consider, e.g., heavy-tailed noise models [31] and adversarially corrupted observations [8]. The second assumption is typically generalized via *heteroscedastic* Gaussian process (GP), allowing an explicit dependence of the noise distribution on the evaluation points [6, 7, 10, 20]. Similarly, in this work, we consider heteroscedastic GP models, but unlike the previous works, we specifically focus on the risk that is associated with querying and reporting noisy points.

Several works have recently considered robust and risk-averse aspects in BO. Their central focus is on designing robust strategies and protecting against the change/shift in uncontrollable covariates. They study various notions including worst-case robustness [9], distributional robustness [19, 30], robust mixed strategies [33] and other notions of risk-aversion [11, 18, 29], and while some of them report robust regret guarantees, their focus is primarily on the robustness in the homoscedastic GP setting. Instead, in our setting, we account for the risk that comes from the realization of random noise with *unknown* distribution. Rather than optimizing the expected performance, in our risk-averse setting, we prefer inputs with *lower variance*. To this end, we incorporate the learning of the noise distribution into the optimization procedure via a *mean-variance* objective. The closest to our setting is risk-aversion with respect to noise in multi-armed bandits [32]. Their approach, however, fails to exploit correlation in rewards among similar arms.

**Contributions.** We propose a novel *Risk-averse Heteroscedastic Bayesian optimization* (RAHBO) approach based on the optimistic principle that trades off the expectation and uncertainty of the mean-variance objective function. We model both the objective and variance as (unknown) functions belonging to RKHS space of functions, and propose a practical risk-aware algorithm in the heteroscedastic GP setting. In our theoretical analysis, we establish rigorous sublinear regret guarantees for our algorithm, and provide a robust reporting rule for applications where only a single solution is required. We demonstrate the effectiveness of RAHBO on synthetic benchmarks, as well as on hyperparameter tuning tasks for the Swiss free-electron laser and a machine learning model.

## 2 Problem setting

Let $\mathcal{X}$ be a given compact set of inputs ($\mathcal{X} \subset \mathbb{R}^d$ for some $d \in \mathbb{N}$). We consider a problem of sequentially interacting with a fixed and unknown objective $f : \mathcal{X} \to \mathbb{R}$. At every round of this procedure, the learner selects an action $x_t \in \mathcal{X}$, and obtains a noisy observation

$$y_t = f(x_t) + \xi(x_t), \tag{1}$$

where $\xi(x_t)$ is zero-mean noise independent across different time steps $t$. In this work, we consider sub-Gaussian heteroscedastic noise that depends on the query location.

**Definition 1.** *A zero-mean real-valued random variable $\xi$ is $\rho$–sub-Gaussian, if there exists variance-proxy $\rho^2$ such that $\forall \lambda \in \mathbb{R}, \ \mathbb{E}[e^{\lambda \xi}] \leq e^{\frac{\lambda^2 \rho^2}{2}}$.*

For a sub-Gaussian $\xi$, its variance $\mathbb{V}ar[\xi]$ lower bounds any valid variance-proxy $\rho$, i.e., $\mathbb{V}ar[\xi] \leq \rho^2$. In case $\mathbb{V}ar[\xi] = \rho^2$ holds, $\xi$ is said to be *strictly $\rho$–sub-Gaussian*. Besides zero-mean Gaussian random variables, most standard symmetric bounded random variables (e.g., Bernoulli, beta, uniform, binomial) are strictly sub-Gaussian (see [2, Proposition 1.1]). Throughout the paper, we consider sub-Gaussian noise, and in Section 3.3, we specialize to the case of strictly sub-Gaussian noise.

**Optimization objective.** Unlike the previous works that mostly focus on sequential optimization of $f$ in the homoscedastic noise case, in this work, we consider the trade-off between risk and return in the heteroscedastic case. While there exist a number of risk-averse objectives, we consider the simple and frequently used mean-variance objective (MV) [32]. Here, the objective value at $x \in \mathcal{X}$ is a trade-off between the (mean) return $f(x)$ and the risk expressed by its variance-proxy $\rho^2(x)$:

$$\text{MV}(x) = f(x) - \alpha \rho^2(x), \tag{2}$$

where $\alpha \geq 0$ is a so-called *coefficient of absolute risk tolerance*. In this work, we assume $\alpha$ is fixed and known to the learner. In the case of $\alpha = 0$, maximizing $\text{MV}(x)$ coincides with the standard BO objective.

**Performance metrics.** We aim to construct a sequence of input evaluations $x_t$ that eventually maximizes the risk-averse objective $\text{MV}(x)$. To assess this convergence, we consider two metrics. The first metric corresponds to the notion of cumulative regret similar to the one used in standard BO and bandits. Here, the learner's goal is to maximize its risk-averse cumulative reward over a time horizon $T$, or equivalently minimize its *risk-averse cumulative regret*:

$$R_T = \sum_{t=1}^{T} \Big[ \text{MV}(x^*) - \text{MV}(x_t) \Big], \tag{3}$$

where $x^* \in \arg\max_{x \in \mathcal{X}} \text{MV}(x)$. A sublinear growth of $R_T$ with $T$ implies vanishing average regret $R_T / T \to 0$ as $T \to \infty$. Intuitively, this implies the existence of some $t$ such that $\text{MV}(x_t)$ is arbitrarily close to the optimal value $\text{MV}(x^*)$.

The second metric is used when the learner seeks to simultaneously minimize the number of expensive function evaluations $T$. Namely, for a given accuracy $\epsilon \geq 0$, we report a single "good" risk-averse point $\hat{x}_T \in \mathcal{X}$ after a total of $T$ rounds, that satisfies:

$$\text{MV}(\hat{x}_T) \geq \text{MV}(x^*) - \epsilon. \tag{4}$$

Both metrics are important for choosing risk-averse solutions and which one is preferred depends on the application at hand. For example, risk-averse cumulative regret $R_T$ might be of a greater interest in online recommendation systems, while reporting a single point with high MV value might be more suitable when tuning machine learning hyperparameters. We consider both performance metrics in our experiments.

**Regularity assumptions.** We consider standard smoothness assumptions [9, 35] when it comes to the unknown function $f : \mathcal{X} \to \mathbb{R}$. In particular, we assume that $f(\cdot)$ belongs to a reproducing kernel Hilbert space (RKHS) $\mathcal{H}_\kappa$ (a space of smooth and real-valued functions defined on $\mathcal{X}$), i.e., $f \in \mathcal{H}_\kappa$, induced by a kernel function $\kappa(\cdot, \cdot)$. We also assume that $\kappa(x, x') \leq 1$ for every $x, x' \in \mathcal{X}$. Moreover, the RKHS norm of $f(\cdot)$ is assumed to be bounded $\|f\|_\kappa \leq \mathcal{B}_f$ for some fixed constant $\mathcal{B}_f > 0$. We assume that the noise $\xi(x)$ is $\rho(x)$–sub-Gaussian with variance-proxy $\rho^2(x)$ uniformly bounded $\rho(x) \in [\varrho, \bar{\varrho}]$ for some constant values $\bar{\varrho} \geq \varrho > 0$.

# 3 Algorithms

We first recall the Gaussian process (GP) based framework for sequential learning of RKHS functions from observations with heteroscedastic noise. Then, in Section 3.2, we consider a simple risk-averse Bayesian optimization problem with *known* variance-proxy, and later on in Section 3.3, we focus on our main problem setting in which the variance-proxy is *unknown*.

## 3.1 Bayesian optimization with heteroscedastic noise

Before addressing the risk-averse objective, we briefly recall the standard GP-UCB algorithm [35] in the setting of heteroscedastic sub-Gaussian noise. The regularity assumptions permit the construction of confidence bounds via GP model. Particularly, to decide which point to query at every round, GP-UCB makes use of the posterior GP mean and variance denoted by $\mu_t(\cdot)$ and $\sigma_t^2(\cdot)$, respectively. They are computed based on the previous measurements $y_{1:t} = [y_1, \ldots, y_t]^\top$ and the given kernel $\kappa(\cdot, \cdot)$ :

$$\mu_t(x) = \kappa_t(x)^T (K_t + \lambda \Sigma_t)^{-1} y_{1:t}, \tag{5}$$

$$\sigma_t^2(x) = \frac{1}{\lambda} \big( \kappa(x, x) - \kappa_t(x)^\top (K_t + \lambda \Sigma_t)^{-1} \kappa_t(x) \big), \tag{6}$$

where $\Sigma_t := \mathrm{diag}(\rho^2(x_1), \ldots, \rho^2(x_t))$, $(K_t)_{i,j} = \kappa(x_i, x_j)$, $\kappa_t(x)^T = [\kappa(x_1, x), \ldots, \kappa(x_t, x)]^T$, $\lambda > 0$ and prior modelling assumptions are $\xi(\cdot) \sim \mathcal{N}(0, \rho^2(\cdot))$ and $f \sim GP(0, \lambda^{-1}\kappa)$.

At time $t$, GP-UCB maximizes the upper confidence bound of $f(\cdot)$, i.e.,

$$x_t \in \underset{x \in \mathcal{X}}{\arg\max} \ \underbrace{\mu_{t-1}(x) + \beta_t \sigma_{t-1}(x)}_{=: \mathrm{ucb}_t^f(x)}. \tag{7}$$

If the noise $\xi_t(x_t)$ is heteroscedastic and $\rho(x_t)$-sub-Gaussian, the following confidence bounds hold:

**Lemma 1** (Lemma 7 in [20]). *Let $f \in \mathcal{H}_\kappa$, and $\mu_t(\cdot)$ and $\sigma_t^2(\cdot)$ be defined as in Eqs. (5) and (6) with $\lambda > 0$. Assume that the observations $(x_t, y_t)_{t \geq 1}$ satisfy Eq. (1). Then the following holds for all $t \geq 1$ and $x \in \mathcal{X}$ with probability at least $1 - \delta$:*

$$|\mu_{t-1}(x) - f(x)| \leq \underbrace{\left( \sqrt{2 \ln \left( \frac{\det(\lambda \Sigma_t + K_t)^{1/2}}{\delta \det(\lambda \Sigma_t)^{1/2}} \right)} + \sqrt{\lambda} \|f\|_\kappa \right)}_{:= \beta_t} \sigma_{t-1}(x). \tag{8}$$

Here, $\beta_t$ stands for the parameter that balances between exploration vs. exploitation and ensures the validity of confidence bounds. The analogous concentration inequalities in case of homoscedastic noise were considered in [1, 14, 35].

**Failure of GP-UCB in the risk-averse setting.** GP-UCB is guaranteed to achieve sublinear cumulative regret with high probability in the risk-neutral (homoscedastic/heteroscedastic) BO setting [14, 35]. However, for the risk-averse setting in Eq. (2), the maximizers $x^* \in \arg\max_{x \in \mathcal{X}} \mathrm{MV}(x)$ and $x_f^* \in \arg\max_{x \in \mathcal{X}} f(x)$ might not coincide, and consequently, $\mathrm{MV}(x^*)$ can be significantly larger than $\mathrm{MV}(x_f^*)$. This is illustrated in Figure 1, where GP-UCB most frequently chooses optimum $A$ of the highest risk.

## 3.2 Warm up: Known variance-proxy

We remedy the previous issue with GP-UCB by proposing a natural *Risk-averse Heteroscedastic* BO (RAHBO) in case of the known variance-proxy $\rho^2(\cdot)$. At each round $t$, RAHBO chooses the action:

$$x_t \in \underset{x \in \mathcal{X}}{\arg\max} \ \mu_{t-1}(x) + \beta_t \sigma_{t-1}(x) - \alpha \rho^2(x), \tag{9}$$

where $\beta_t$ is from Lemma 1 and $\alpha$ is from Eq. (2). In the next section, we further relax the assumption of the variance-proxy and consider a more practical setting when $\rho^2(\cdot)$ is unknown to the learner. For the current setting, the performance of RAHBO is formally captured in the following proposition.

**Proposition 1.** *Consider any $f \in \mathcal{H}_\kappa$ with $\|f\|_\kappa \leq \mathcal{B}_f$ and sampling model from Eq. (1) with known variance-proxy $\rho^2(x)$. Let $\{\beta_t\}_{t=1}^T$ be set as in Lemma 1 with $\lambda = 1$. Then, with probability at least $1 - \delta$, RAHBO attains cumulative risk-averse regret $R_T = \mathcal{O}\big( \beta_T \sqrt{T \gamma_T (\bar{\varrho}^2 + 1)} \big)$.*

Here, $\gamma_T$ denotes the *maximum information gain* [35] at time $T$ defined via mutual information $I(y_{1:T}, f_{1:T})$ between evaluations $y_{1:T}$ and $f_{1:T} = [f(x_1), \ldots, f(x_T)]^\top$ at points $A \subset D$:

$$\gamma_T := \max_{A \subset \mathcal{X}, \, |A|=T} I(y_{1:T}, f_{1:T}), \tag{10}$$

$$\text{where} \quad I(y_{1:T}, f_{1:T}) = \frac{1}{2} \sum_{t=1}^{T} \ln\left(1 + \frac{\sigma_{t-1}^2(x_t)}{\rho^2(x_t)}\right) \tag{11}$$

in case of heteroscedastic noise (see Appendix A.1.1). The upper bounds on $\gamma_T$ are provided in [35] widely used kernels. These upper bounds typically scale sublinearly in $T$; for linear kernel $\gamma_T = \mathcal{O}(d \log T)$, and in case of squared exponential kernel $\gamma_T = \mathcal{O}(d(\log T)^{d+1})$. While these bounds are derived assuming the homoscedastic GP setting with some fixed constant noise variance, we show (in Appendix A.1.3) that the same rates (up to a multiplicative constant factor) apply in the heteroscedastic case.

### 3.3 RAHBO for unknown variance-proxy

In the case of unknown variance-proxy, the confidence bounds for the unknown $f(x)$ in Lemma 1 can not be readily used, and we construct new ones on the combined mean-variance objective. To learn about the unknown $\rho^2(x)$, we make some further assumptions.

**Assumption 1.** *The variance-proxy $\rho^2(x)$ belongs to an RKHS induced by some kernel $\kappa^{var}$, i.e., $\rho^2 \in \mathcal{H}_{\kappa^{var}}$, and its RKHS norm is bounded $\|\rho^2\|_{\kappa^{var}} \leq \mathcal{B}_{var}$ for some finite $\mathcal{B}_{var} > 0$. Moreover, the noise $\xi(x)$ in Eq. (1) is strictly $\rho(x)$–sub-Gaussian, i.e., $\mathbb{V}ar[\xi(x)] = \rho^2(x)$ for every $x \in \mathcal{X}$.*

As a consequence of our previous assumption, we can now focus on estimating the variance since $\mathbb{V}ar[\xi(\cdot)]$ and $\rho^2(\cdot)$ coincide. In particular, to estimate $\mathbb{V}ar[\xi(\cdot)]$ we consider a *repeated experiment setting*, where for each $x_t$ we collect $k > 1$ evaluations $\{y_i(x_t)\}_{i=1}^k$, $y_i(x_t) = f(x_t) + \xi_i(x_t)$. Then, the sample mean and variance of $\xi(x_t)$ are given as:

$$\hat{m}_k(x_t) = \frac{1}{k} \sum_{i=1}^{k} y_i(x_t) \quad \text{and} \quad \hat{s}_k^2(x_t) = \frac{1}{k-1} \sum_{i=1}^{k} \left(y_i(x_t) - \hat{m}_k(x_t)\right)^2. \tag{12}$$

The key idea is that for strictly sub-Gaussian noise $\xi(x)$, $\hat{s}_{1:t}^2 = [\hat{s}_k^2(x_1), \ldots, \hat{s}_k^2(x_t)]^\top$ yields *unbiased, but noisy* evaluations of the unknown variance-proxy $\rho_{1:t}^2 = [\rho^2(x_1), \ldots, \rho^2(x_t)]^\top$, i.e.,

$$\hat{s}_k^2(x_t) = \rho^2(x_t) + \eta(x_t) \tag{13}$$

with zero-mean noise $\eta(x_t)$. In order to efficiently estimate $\rho^2(\cdot)$, we need an additional assumption.

**Assumption 2.** *The noise $\eta(x)$ in Eq. (13) is $\rho_\eta(x)$–sub-Gaussian with known $\rho_\eta^2(x)$ and the realizations $\{\eta(x_t)\}_{t \geq 1}$ are independent between $t$.*

We note that a similar assumption is made in [32] in the multi-armed bandit setting. The fact that $\rho_\eta^2(\cdot)$ is known is rather mild as Assumption 1 allows controlling its value. For example, in case of strictly sub-Gaussian $\eta(x)$ we show (in Appendix A.2) that $\mathbb{V}ar[\eta(\cdot)] = \rho_\eta^2(\cdot) \leq 2\rho^4(\cdot)/(k-1)$. Then, given that $\rho^2(\cdot) \leq \bar{\varrho}^2$, we can utilize the following (rather conservative) bound as a variance-proxy, i.e., $\rho_\eta^2(x) = 2\bar{\varrho}^4/(k-1)$.

**RAHBO algorithm.** We present our Risk-averse Heteroscedastic BO approach for unknown variance-proxy in Algorithm 1. Our method relies on building the following two GP models.

Firstly, we use sample variance evaluations $\hat{s}_{1:t}^2$ to construct a GP model for $\rho^2(\cdot)$. The corresponding $\mu_{t-1}^{var}(\cdot)$ and $\sigma_{t-1}^{var}(\cdot)$ are computed as in Eqs. (5) and (6) by using kernel $\kappa^{var}$, variance-proxy $\rho_\eta^2(\cdot)$ and noisy observations $\hat{s}_{1:t}^2$. Consequently, we build the upper and lower confidence bounds $\text{ucb}_t^{var}(\cdot)$ and $\text{lcb}_t^{var}(\cdot)$ of the variance-proxy $\rho^2(\cdot)$ and we set $\beta_t^{var}$ according to Lemma 1:

$$\text{ucb}_t^{var}(x) := \mu_{t-1}^{var}(x) + \beta_t^{var} \sigma_{t-1}^{var}(x), \tag{14}$$

$$\text{lcb}_t^{var}(x) := \mu_{t-1}^{var}(x) - \beta_t^{var} \sigma_{t-1}^{var}(x). \tag{15}$$

Secondly, we use sample mean evaluations $\hat{m}_{1:t} = [\hat{m}_k(x_1), \ldots, \hat{m}_k(x_t)]^\top$ to construct a GP model for $f(\cdot)$. The mean $\mu_t(\cdot)$ and variance $\sigma_t^2(\cdot)$ in Eqs. (5) and (6), however, rely on the unknown

---

**Algorithm 1** Risk-averse Heteroscedastic Bayesian Optimization (RAHBO)

---

**Require:** Parameters $\alpha, \{\beta_t, \beta_t^{var}\}_{t \geq 1}, \lambda, k$, Prior $\mu_0^f = \mu_0^{var} = 0$, Kernel functions $\kappa, \kappa^{var}$

1: **for** $t = 1, 2, \ldots$ **do**
2:    Construct confidence bounds $\mathrm{ucb}_t^{var}(\cdot)$ and $\mathrm{lcb}_t^{var}(\cdot)$ as in Eqs. (14) and (15)
3:    Construct $\mathrm{ucb}_t^f(\cdot)$ as in Eq. (7)
4:    Select $x_t \in \arg\max_{x \in \mathcal{X}} \mathrm{ucb}_t^f(x) - \alpha\,\mathrm{lcb}_t^{var}(x)$
5:    Observe $k$ samples: $y_i(x_t) = f(x_t) + \xi_i(x_t)$   for every   $i \in [k]$
6:    Use samples $\{y_i(x_t)\}_{i=1}^k$ to compute sample mean $\hat{m}_k(x_t)$ and variance $\hat{s}_k^2(x_t)$ as in Eq. (12)
7:    Use $x_t, \hat{s}_k^2(x_t)$ to update posterior $\mu_t^{var}(\cdot)$ and $\sigma_t^{var}(\cdot)$ as in Eqs. (32) and (33)
8:    Use $\mathrm{ucb}_t^{var}(\cdot)$ to compute $\hat{\Sigma}_t$ as in Eq. (16)
9:    Use $x_t, \hat{m}_k(x_t)$ and $\hat{\Sigma}_t$ to update posterior $\mu_t(\cdot)$ and $\sigma_t(\cdot)$ as in Eqs. (5) and (6)
10: **end for**

---

variance-proxy $\rho^2(\cdot)$ in $\Sigma_t$, an we thus use its upper confidence bound $\mathrm{ucb}_t^{var}(\cdot)$ truncated with $\bar{\varrho}^2$:

$$\hat{\Sigma}_t := \tfrac{1}{k}\mathrm{diag}\big(\min\{\mathrm{ucb}_t^{var}(x_1), \bar{\varrho}^2\}, \ldots, \min\{\mathrm{ucb}_t^{var}(x_t), \bar{\varrho}^2\}\big), \tag{16}$$

where $\hat{\Sigma}_t$ is corrected by $k$ since every evaluation in $\hat{m}_{1:t}$ is an average over $k$ samples. This substitution of the unknown variance-proxy by its conservative estimate guarantees that the confidence bounds $\mathrm{ucb}_t^f(x) := \mu_{t-1}(x) + \beta_t \sigma_{t-1}(x)$ on $f$ also hold with high probability (conditioning on the confidence bounds for $\rho(\cdot)$ holding true; see Appendix A.3 for more details).

Finally, we define the acquisition function as $\mathrm{ucb}_t^{MV}(x) := \mathrm{ucb}_t^f(x) - \alpha\mathrm{lcb}_t^{var}(x)$, i.e., selecting $x_t \in \arg\max_{x \in \mathcal{X}} \mathrm{ucb}_t^{MV}(x)$ at each round $t$.

The proposed algorithm leads to new maximum information gains $\hat{\gamma}_T = \max_A I(\hat{m}_{1:T}, f_{1:T})$ and $\Gamma_T = \max_A I(\hat{s}_{1:T}^2, \rho_{1:T}^2)$ for sample mean $\hat{m}_{1:T}$ and sample variance $\hat{s}_{1:T}^2$ evaluations. The corresponding mutual information in $\hat{\gamma}_T$ and $\Gamma_T$ is computed according to Eq. (11) for heteroscedastic noise with variance-proxy $\bar{\varrho}^2/k$ and $\rho_\eta^2$, respectively (see Appendix A.4). The performance of RAHBO is captured in the following theorem.

**Theorem 1.** *Consider any $f \in \mathcal{H}_\kappa$ with $\|f\|_\kappa \leq \mathcal{B}_f$ and sampling model in Eq. (1) with unknown variance-proxy $\rho^2(x)$ that satisfies Assumptions 1 and 2. Let $\{x_t\}_{t=1}^T$ denote the set of actions chosen by* RAHBO *(Algorithm 1) over $T$ rounds. Set $\{\beta_t\}_{t=1}^T$ and $\{\beta_t^{var}\}_{t=1}^T$ according to Lemma 1 with $\lambda = 1$, $\mathcal{R}^2 = \max_{x \in \mathcal{X}} \rho_\eta^2(x)$ and $\rho(\cdot) \in [\varrho, \bar{\varrho}]$. Then, the risk-averse cumulative regret $R_T$ of* RAHBO *is bounded as follows:*

$$\Pr\left\{R_T \leq \beta_T k\sqrt{\frac{2T\hat{\gamma}_T}{\ln(1 + k/\bar{\varrho}^2)}} + \alpha\beta_T^{var}k\sqrt{\frac{2T\Gamma_T}{\ln(1 + \mathcal{R}^{-2})}}, \quad \forall T \geq 1\right\} \geq 1 - \delta. \tag{17}$$

The risk-averse cumulative regret of RAHBO depends sublinearly on $T$ for most of the popularly used kernels. This follows from the implicit sublinear dependence on $T$ in $\beta_T, \beta_T^{var}$ and $\hat{\gamma}_T, \Gamma_T$ (the bounds in case of heteroscedastic noise replicate the ones used in Proposition 1 as shown in Appendices A.1.2 and A.1.3). Finally, the result of Theorem 1 provides a non-trivial trade-off for number of repetitions $k$ where larger $k$ increases sample complexity but also leads to better estimation of the noise model. Furthermore, we obtain the bound for the number of rounds $T$ required for identifying an $\epsilon$-optimal point:

**Corollary 1.1.** *Consider the setup of Theorem 1. Let $A = \{x_t\}_{t=1}^T$ denote actions selected by* RAHBO *over $T$ rounds. Then, with probability at least $1 - \delta$, the reported point $\hat{x}_T := \arg\max_{x_t \in A} \mathrm{lcb}_t^{MV}(x_t)$, where $\mathrm{lcb}_t^{MV}(x_t) = \mathrm{lcb}_t^f(x) - \alpha\,\mathrm{ucb}_t^{var}(x)$, achieves $\epsilon$-accuracy, i.e., $MV(x^*) - MV(\hat{x}_T) \leq \epsilon$, after $T \geq \frac{4\beta_T^2\hat{\gamma}_T/\ln(1+k/\bar{\varrho}^2) + 4\alpha(\beta_t^{var})^2\Gamma_T/\ln(1+\mathcal{R}^{-2})}{\epsilon^2}$ rounds.*

The previous result demonstrates the sample complexity rates when a single risk-averse reported solution is required. We note that both Theorem 1 and Corollary 1.1 provide guarantees for choosing risk-averse solutions, and depending on application at hand, we might consider either one of the proposed performance metrics. We demonstrate use-cases for both in the following section.

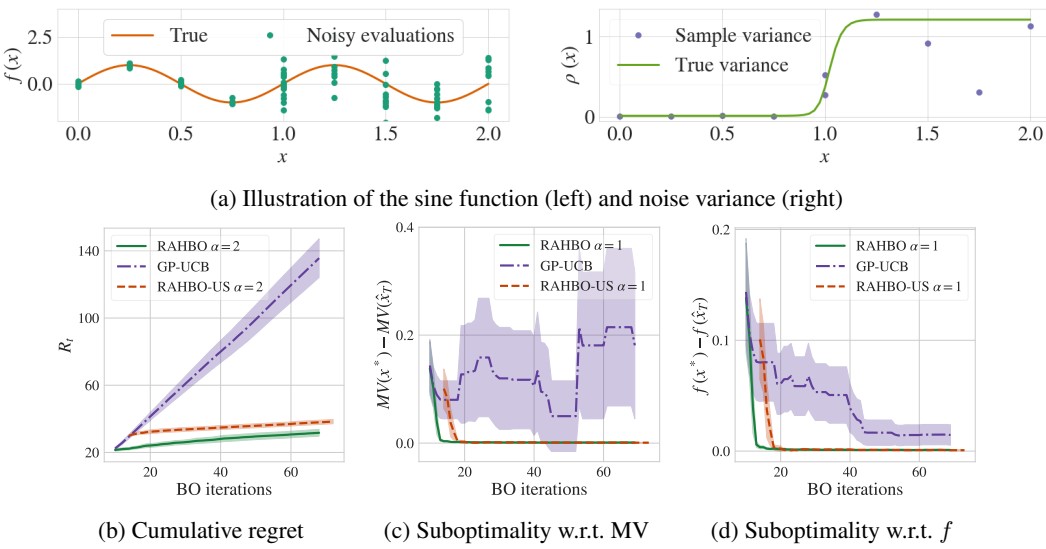

(a) Illustration of the sine function (left) and noise variance (right)

(b) Cumulative regret      (c) Suboptimality w.r.t. MV      (d) Suboptimality w.r.t. $f$

Figure 2: (a) Unknown true objective along with noisy evaluations with varying noise level (left) and unknown true noise variance and its evaluations (right). (b) Cumulative regret. (c) Simple MV regret for reporting rule $\hat{x}_T = \arg\max_{x_t} \mathrm{lcb}_T(x_t)$. (c) Simple regret $f(x^*) - f(\hat{x}_T)$ for the unknown function at the reported point $\hat{x}_T$ from (d). RAHBO not only leads to strong results in terms of MV but also in terms of the mean objective $f(x)$.

## 4 Experiments

In this section, we experimentally validate RAHBO on two synthetic examples and two real hyperparameter tuning tasks, and compare it with the baselines. We provide an open-source implementation of our method.[1]

**Baselines.** We compare against two baselines: As the first baseline, we use GP-UCB with heteroscedastic noise as a standard risk-neutral algorithm that optimizes the unknown $f(x)$. As the second one, we consider a risk-averse baseline that uniformly learns variance-proxy $\rho^2(x)$ *before* the optimization procedure, in contrast to RAHBO which learns the variance-proxy on the fly. We call it RAHBO-US, standing for RAHBO with uncertainty sampling. It consists of two stages: (i) uniformly learning $\rho^2(x)$ via uncertainty sampling, (ii) GP-UCB applied to the mean-variance objective, in which instead of the unknown $\rho^2(x)$ we use the mean of the learned model. Note that RAHBO-US is the closest to the contextual BO setting [18], where the context distribution is assumed to be known.

**Experimental setup.** At each iteration $t$, an algorithm queries a point $x_t$ and observes sample mean and sample variance of $k$ observations $\{y_i(x_t)\}_{i=1}^{k}$. We use a heteroscedastic GP for modelling $f(x)$ and a homoscedastic GP for $\rho^2(x)$. We set $\lambda = 1$ and $\beta_t = 2$, which is commonly used in practice to improve performance over the theoretical results. Before the BO procedure, we determine the GP hyperparameters maximizing the marginal likelihood. To this end, we use initial points that are same for all the baselines and are chosen via Sobol sequence that generates low discrepancy quasi-random samples. We repeat each experiment several times, generating new initial points for every repetition. We use two metrics: (a) risk-averse cumulative regret $R_t$ computed for the acquired inputs; (b) simple regret $\mathrm{MV}(x^*) - \mathrm{MV}(\hat{x}_T)$ computed for inputs as reported via Corollary 1.1. For each metric, we report its mean $\pm$ two standard errors over the repetitions.

**Example function** We first illustrate the methods performance on a sine function depicted in Figure 2a. This function has two global optimizers. We induce a heteroscedastic zero-mean Gaussian noise on the measurements. We use a sigmoid function for the noise variance, as depicted in Figure 2a, that induces small noise on $[0, 1]$ and higher noise on $(1, 2]$. We initialize the algorithms by selecting 10 inputs $x$ at random and keep these points the same for all the algorithms. We use $k = 10$ samples at each chosen $x_t$. The number of acquisition rounds is $T = 60$. We repeat the experiment 30 times for each method and show their average performances in Figure 2.

---

[1]`https://github.com/Avidereta/risk-averse-hetero-bo`

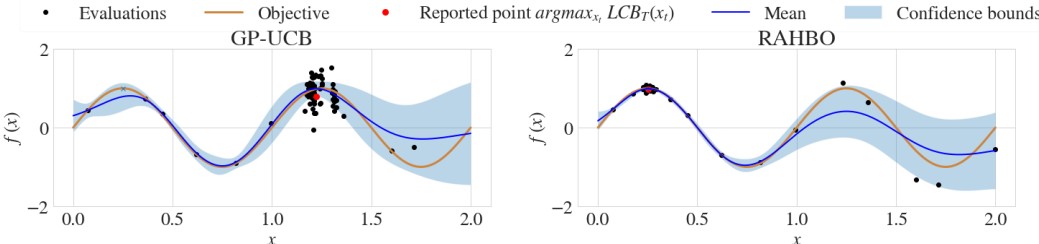

Figure 3: GP models fitted for GP-UCB (left) and RAHBO (right) for sine function. After initialization with the same sampled points, GP-UCB concentrates on the high-noise region whereas RAHBO prefers small variance. Additional plots are presented in Appendix A.6.

**Branin benchmark** Next, we evaluate the methods on the (negated) Branin benchmark function in Figure 1a, achieving its optimum value $f^* = -0.4$ at $(-\pi, 12.3), (\pi, 2.3), (9.4, 2.5)$. The heteroscedastic variance function illustrated in Figure 1b defines different noise variances for the three optima. We initialize all algorithms by selecting 10 inputs. We use $k = 10$ samples to estimate the noise variance. The number of acquisition rounds is $T = 150$. We repeat BO 25 times and show the results in Figures 1c and 5a. Figure 1c provides more intuition behind the observed regret: UCB exploits the noisiest maxima the most, while RAHBO prefers smaller variance.

**Tuning Swiss free-electron laser** In this experiment, we tune the parameters of Swiss X-ray free-electron laser (SwissFEL), an important scientific instrument that generates very short pulses of X-ray light and enables researchers to observe extremely fast processes. The main objective is to maximize the pulse energy measured by a gas detector, that is a time-consuming and repetitive task during the SwissFEL operation. Such (re-)tuning takes place while user experiments on SwissFEL are running, and thus the cumulative regret is the metric of high importance in this application.

We use real SwissFEL measurements collected in [21] to train a neural network surrogate model, and use it to simulate the SwissFEL objective $f(x)$ for new parameter settings $x$. We similarly fit a model of the heteroscedastic variance by regressing the squared residuals via a GP model. Here, we focus on the calibration of the four most sensitive parameters.

We report our comparison in Figure 4 where we also assess the effect of varying the coefficient of absolute risk tolerance $\alpha$. We use 30 points to initialize the baselines and then perform 200 acquisition rounds. We repeat each experiment 15 times. In Figure 4a we plot the empirical frequency of the true (unknown to the methods) values $f(x_t)$ and $\rho^2(x_t)$ at the inputs $x_t$ acquired by the methods. The empirical frequency for $\rho^2(x)$ illustrates the tendency of risk-neutral GP-UCB to query points with higher noise, while risk-averse achieves substantially reduced variance and minimal reduction in mean performance. Sometimes, risk-neutral GP-UCB also fails to succeed in querying points with the highest $f$-value. That tendency results in lower cumulative regret for RAHBO in Figures 4c and 4d. We also compare the performance of the reporting rule from Corollary 1.1 in Figure 4b, where we plot error bars with standard deviation both for $f(\hat{x}_T)$ and $\rho^2(\hat{x}_T)$ at the reported point $\hat{x}_T$. As before, RAHBO drastically reduces the variance compared to GP-UCB, while having only slightly lower mean performance. Additional results are presented in Figure 10 in Appendix.

**Random Forest tuning** BO is widely used by cloud services for tuning machine learning hyperparameters and the resulting models might be then used in high-stakes applications such as credit scoring or fraud detection. In k-fold cross-validation, the average metric over the validation sets is optimized – a canonical example of the *repeated experiment setting* that we consider in the paper. High across-folds variance is a practical problem [24] where the mean-variance approach might be beneficial.

In our experiment, we tune hyperparameters of a random forest classifier (RF) on a dataset of fraudulent credit card transactions [23].[2] It consist of 285k transactions with 29 features (processed due to confidentiality issues) that are distributed over time, and only 0.2% are fraud examples (see Appendix for more details). The search space for the RF hyperparameters is also provided in the Appendix. We use the balanced accuracy score and 5 validation folds, i.e., $k = 5$, and each validation fold is shifted in time with respect to the training data. We seek not only for high performance *on average* but also for low variance across the validation folds that have different time shifts with respect to the training data.

---

[2]`https://www.kaggle.com/mlg-ulb/creditcardfraud`

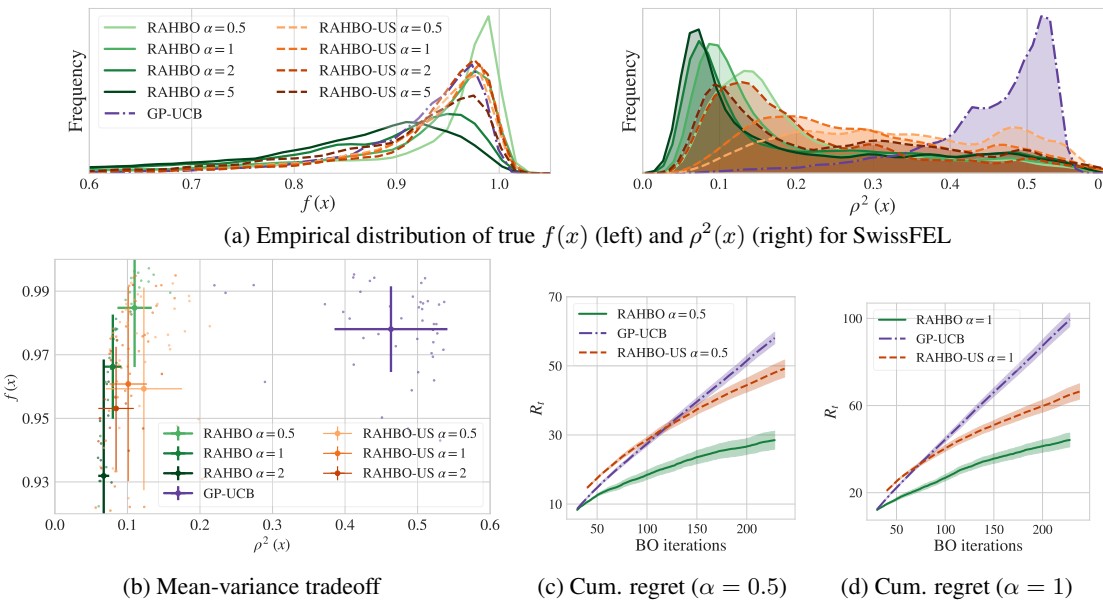

(a) Empirical distribution of true $f(x)$ (left) and $\rho^2(x)$ (right) for SwissFEL

(b) Mean-variance tradeoff      (c) Cum. regret ($\alpha = 0.5$)      (d) Cum. regret ($\alpha = 1$)

Figure 4: Experimental results for SwissFEL: **(a)** Distributions of $f(x)$ and $\rho^2(x)$ for *all points* queried during the optimization. GP-UCB queries points with higher noise (but not necessarily high return $f$) in contrast to the risk-averse methods. **(b)** Mean $f(\hat{x}_T)$ and variance $\rho^2(\hat{x}_T)$ at the *reported* $\hat{x}_T = \arg\max_{x_t} \mathrm{lcb}_T(x_t)$: for each method, we repeat BO experiment 15 times (separate points) and plot corresponding standard deviation error bars. RAHBO reports solutions with reasonable mean-variance tradeoff, while GP-UCB produces solutions with high mean value but also high noise variance. **(c-d)** Cum. regret for $\alpha = 0.5$ and $\alpha = 1$ (see more in Appendix A.7.1).

We initialize the algorithms by selecting 10 hyperparameter settings and keep these points the same for all algorithms. We use Matérn 5/2 kernels with Automatic Relevance Discovery (ARD) and normalize the input features to the unit cube. The number of acquisition rounds in one experiment is 50 and we repeat each experiment 15 times. We demonstrate our results in Figures 5b and 5c where we plot mean $\pm$ 2 standard errors. While both RAHBO and GP-UCB perform comparable in terms of the mean error, its standard deviation for RAHBO is smaller.

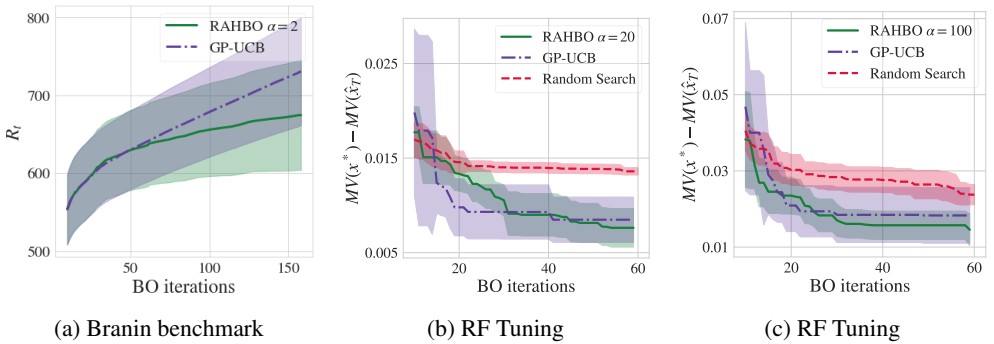

(a) Branin benchmark      (b) RF Tuning      (c) RF Tuning

Figure 5: **Branin**: (a) Cumulative regret. **Random Forest**: (b-c) Simple regret for the reported $\hat{x}_T = \arg\max_{x_t} MV(x_t)$ for (b) $\alpha = 20$ and (c) $\alpha = 100$. While both methods have comparable mean, RAHBO has consistently lower variance.

## 5 Conclusion

In this work, we generalize Bayesian optimization to the risk-averse setting and propose RAHBO algorithm aiming to find an input with both large expected return and small input-dependent noise variance. Both the mean objective and the variance are assumed to be unknown a priori and hence are estimated online. RAHBO is equipped with theoretical guarantees showing (under reasonable

assumptions) sublinear dependence on the number of evaluation rounds $T$ both for cumulative risk-averse regret and $\epsilon$-accurate mean-variance metric. The empirical evaluation of the algorithm on synthetic benchmarks and hyperparameter tuning tasks demonstrate promising examples of heteroscedastic use-cases benefiting from RAHBO.

### Acknowledgements

This research has been gratefully supported by NCCR Automation grant 51NF40 180545, by ERC under the European Union's Horizon grant 815943, SNSF grant 200021_172781, and ETH Zürich Postdoctoral Fellowship 19-2 FEL-47. The authors thank Sebastian Curi, Mojmír Mutný and Johannes Kirschner as well as the anonymous reviewers of this paper for their helpful feedback.

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
