# A Appendix
## Risk-averse Heteroscedastic Bayesian Optimization

(Anastasiia Makarova, Ilnura Usmanova, Ilija Bogunovic, Andreas Krause)

### A.1 Details on Proposition 1

We first provide the proof of Proposition 1 for cumulative risk-averse regret Eq. (3) with known variance-proxy $\rho^2(\cdot)$ (see Definition 1) (Appendix A.1.1). We further provide data-independent bounds for $\beta_T$ (Appendix A.1.2) and maximum information gain $\gamma_T$ (Appendix A.1.3) that together conclude the proof for sub-linear on $T$ regret guarantees for most of the popularly used kernels.

### A.1.1 Proof Proposition 1

**Proposition 1.** *Consider any $f \in \mathcal{H}_\kappa$ with $\|f\|_\kappa \leq \mathcal{B}_f$ and sampling model from Eq.* (1) *with known variance-proxy $\rho^2(x)$. Let $\{\beta_t\}_{t=1}^T$ be set as in Lemma 1 with $\lambda = 1$. Then, with probability at least $1 - \delta$, RAHBO attains cumulative risk-averse regret $R_T = \mathcal{O}\big(\beta_T \sqrt{T\gamma_T(\bar{\varrho}^2 + 1)}\big)$.*

*Proof.* The main steps of the proof are as follows: In *Step 1*, we derive the upper and the lower confidence bounds, $\mathrm{ucb}_t^{\mathrm{MV}}(x_t)$ and $\mathrm{lcb}_t^{\mathrm{MV}}(x_t)$, on $\mathrm{MV}(x_t)$ at iteration $t$. In *Step 2*, we bound the instantaneous risk-averse regret $r(x_t) := \mathrm{MV}(x^*) - \mathrm{MV}(x_t)$. In *Step 3*, we derive mutual information $I(y_{1:T}, f_{1:T})$ in case of the heteroscedastic noise. In *Step 4*, we bound the sum of variances $\sum_{t=1}^T \sigma_{t-1}(x_t)$ via mutual information $I(y_{1:T}, f_{1:T})$. In *Step 5*, we bound the cumulative regret $R_T = \sum_{t=1}^T r(x_t)$ based on the previous steps.

*Step 1*: *On the confidence bounds for MV(x).*
In case of known variance-proxy $\rho^2(x)$, the confidence bounds for $\mathrm{MV}(x)$ at iteration $t$ can be directly obtained based on the posterior $\mu_t(x)$ and $\sigma_t(x)$ for $f(x)$ defined in Eqs. (5) and (6). Particularly, for $\beta_t = \beta_t(\delta)$ defined in Eq. (8), $\Pr\left\{ \mathrm{lcb}_t^{\mathrm{MV}}(x) \leq \mathrm{MV}(x) \leq \mathrm{ucb}_t^{\mathrm{MV}}(x) \; \forall x \in \mathcal{X}, \forall t \geq 0 \right\} \geq 1 - \delta$ with the confidence bounds:

$$\mathrm{lcb}_t^{\mathrm{MV}}(x) := \mu_{t-1}(x) - \beta_t \sigma_{t-1}(x) - \alpha\rho^2(x), \tag{18}$$

$$\mathrm{ucb}_t^{\mathrm{MV}}(x) := \mu_{t-1}(x) + \beta_t \sigma_{t-1}(x) - \alpha\rho^2(x). \tag{19}$$

*Step 2*: *On bounding the instantaneous risk-averse regret $r_t(x)$.* We have

$$\begin{aligned}
r(x_t) &= \mathrm{MV}(x^*) - \mathrm{MV}(x_t) \\
&\leq \mathrm{ucb}_t^{\mathrm{MV}}(x^*) - \mathrm{lcb}_t^{\mathrm{MV}}(x_t) \\
&\leq \mathrm{ucb}_t^{\mathrm{MV}}(x_t) - \mathrm{lcb}_t^{\mathrm{MV}}(x_t) = 2\beta_t \sigma_{t-1}(x_t),
\end{aligned}$$

where the first inequality is due to the definition of confidence bounds, the second is due to the acquisition strategy $x_t \in \arg\max_{x \in \mathcal{X}} \mathrm{ucb}_t^{\mathrm{MV}}(x)$; and the equality further expands $\mathrm{lcb}_t^{\mathrm{MV}}(x)$ and $\mathrm{ucb}_t^{\mathrm{MV}}(x)$. Thus, the cumulative regret can be bounded as follows:

$$R_T = \sum_{t=1}^T r(x_t) \leq \sum_{t=1}^T 2\beta_t \sigma_{t-1}(x_t) \leq 2\beta_T \sum_{t=1}^T \sigma_{t-1}(x_t), \tag{20}$$

where the last inequality holds since $\{\beta_t\}_{t=1}^T$ is a non-decreasing sequence.

*Step 3*: *On mutual information $I(y_{1:T}, f_{1:T})$ and maximum information gain $\gamma_T$.*
Mutual information $I(y_{1:T}, f_{1:T})$ between the vector of evaluations $y_{1:T} \in \mathbb{R}^T$ at points $A = \{x_t\}_{t=1}^T$ and $f_{1:T} = [f(x_1), \ldots, f(x_T)]^\top$ is defined by

$$I(y_{1:T}, f_{1:T}) = H(y_{1:T}) - H(y_{1:T}|f_{1:T}),$$

where $H(\cdot)$ denotes entropy. Under the modelling assumptions $f_{1:T} \sim \mathcal{N}(0, \lambda^{-1}K_T)$ and $\xi_{1:T} \sim \mathcal{N}(0, \Sigma_T)$ for the noise $\xi_{1:T} = [\xi(x_1), \ldots, \xi(x_T)]^\top$, the measurements are distributed as $y_{1:T} \sim \mathcal{N}(0, \lambda^{-1}K_T + \Sigma_T)$ and $y_t|y_{1:t-1} \sim \mathcal{N}(\mu_{t-1}(x_t), \rho^2(x_t) + \sigma_{t-1}^2(x_t))$, where $\sigma_{t-1}^2(\cdot)$ is defined in

Eq. (6). Hence, the entropy of each new measurement $y_t$ conditioned on the previous history $y_{1:t-1}$ is:

$$H(y_t|y_{1:t-1}) = \frac{1}{2}\ln\left(2\pi e\left(\rho^2(x_t) + \sigma_{t-1}^2(x_t)\right)\right)$$

$$= \frac{1}{2}\ln\left(2\pi e\rho^2(x_t)\left(1 + \frac{\sigma_{t-1}^2(x_t)}{\rho^2(x_t)}\right)\right)$$

$$= \frac{1}{2}\ln\left(2\pi e\rho^2(x_t)\right) + \frac{1}{2}\ln\left(1 + \frac{\sigma_{t-1}^2(x_t)}{\rho^2(x_t)}\right),$$

$$H(y_{1:T}) = \sum_{t=1}^{T} H(y_t|y_{1:t-1}) = \frac{1}{2}\sum_{t=1}^{T}\ln\left(2\pi e\rho^2(x_t)\right) + \frac{1}{2}\sum_{t=1}^{T}\ln\left(1 + \frac{\sigma_{t-1}^2(x_t)}{\rho^2(x_t)}\right),$$

$$H(y_{1:T}|f_{1:T}) = \sum_{t=1}^{T} H(y_t|f_t) = \frac{1}{2}\sum_{t=1}^{T}\ln(2\pi e\rho^2(x_t)).$$

Therefore, the information gain for $y_{1:T}$ is:

$$I(y_{1:T}, f_{1:T}) = H(y_{1:T}) - H(y_{1:T}|f_{1:T}) = \frac{1}{2}\sum_{t=1}^{T}\ln\left(1 + \frac{\sigma_{t-1}^2(x_t)}{\rho^2(x_t)}\right). \tag{21}$$

Then, by definition of maximum information gain:

$$\gamma_T := \max_{A \subset \mathcal{X},\, |A|=T} I(y_{1:T}, f_{1:T}) \geq \frac{1}{2}\sum_{t=1}^{T}\ln\left(1 + \frac{\sigma_{t-1}^2(x_t)}{\rho^2(x_t)}\right). \tag{22}$$

**Step 4**: *On bounding $\sum_{t=1}^{T}\sigma_{t-1}(x_t)$.*

$$\sum_{t=1}^{T}\sigma_{t-1}(x_t) = \sum_{t=1}^{T}\frac{\rho(x_t)}{\rho(x_t)}\sigma_{t-1}(x_t) \leq \sqrt{T\sum_{t=1}^{T}\rho^2(x_t)\frac{\sigma_{t-1}^2(x_t)}{\rho^2(x_t)}}$$

$$\leq \sqrt{T\sum_{t=1}^{T}\frac{1}{\ln(1+\rho^{-2}(x_t))}\ln\left(1 + \frac{\sigma_{t-1}^2(x_t)}{\rho^2(x_t)}\right)}$$

$$\leq \sqrt{\frac{2T}{\ln(1+\bar{\varrho}^{-2})}\underbrace{\frac{1}{2}\sum_{t=1}^{T}\ln\left(1 + \frac{\sigma_{t-1}^2(x_t)}{\rho^2(x_t)}\right)}_{\text{mutual information Eq. (21)}}}, \tag{23}$$

where the first inequality follows from the Cauchy-Schwarz inequality. The second one is due to the fact that for any $s^2 \in [0, \rho^{-2}(x_t)]$ we can bound $s^2 \leq \frac{\rho^{-2}(x_t)}{\ln(1+\rho^{-2}(x_t))}\ln(1+s^2)$, that also holds for $s^2 := \rho^{-2}(x_t)\sigma_{t-1}^2(x_t)$ since $\rho^{-2}(x_t)\sigma_{t-1}^2(x_t) \leq \rho^{-2}(x_t)\lambda^{-1}\kappa(x_t, x_t) \leq \rho^{-2}(x_t)$ for $\lambda \geq 1$. The third inequality is due to $\rho(x) \in [\varrho, \bar{\varrho}]$.

**Step 5**: *Bounding risk-averse cumulative regret $R_T = \sum_{t=1}^{T} r(x_t)$.*
Combining the previous three steps together: Eq. (20), Eq. (22), and Eq. (23) we finally obtain:

$$R_T \leq \sum_{t=1}^{T} 2\beta_t\sigma_{t-1}(x_t) \leq 2\beta_T\sum_{t=1}^{T}\sigma_{t-1}(x_t) \leq 2\beta_T\sqrt{\frac{2T}{\ln(1+\bar{\varrho}^{-2})}\gamma_T}$$

Also, note that for any $\alpha \geq 0$ the bound $\ln(1+\alpha) \geq \frac{\alpha}{1+\alpha}$ holds, thus $\frac{1}{\ln(1+\bar{\varrho}^{-2})} \leq \frac{1+\bar{\varrho}^{-2}}{\bar{\varrho}^{-2}} = \bar{\varrho}^2 + 1$.
Therefore, the cumulative regret can be also bounded as $R_T = O(\beta_T\sqrt{T\gamma_T(\bar{\varrho}^2 + 1)})$. □

### A.1.2 Bounds for $\beta_T$

We provide the bounds for the data-dependent $\beta_T$ that appear in the regret bound (see Eq. (8)). Following our modelling assumptions $f_{1:T} \sim \mathcal{N}(0, \lambda^{-1} K_T)$ and $\xi_{1:T} \sim \mathcal{N}(0, \Sigma_T)$, the information gain $I(y_{1:T}, f_{1:T}) = H(y_{1:T}) - H(y_{1:T}|f_{1:T})$ is given as follows:

$$I(y_{1:T}, f_{1:T}) = \underbrace{\frac{1}{2} \ln \left( \det(2\pi e (\lambda^{-1} K_T + \Sigma_T)) \right)}_{H(y_{1:T})} - \underbrace{\frac{1}{2} \ln \left( \det(2\pi e \Sigma_T) \right)}_{H(y_{1:T}|f_{1:T})} = \frac{1}{2} \ln \left( \frac{\det(K_T + \lambda \Sigma_T)}{\det(\lambda \Sigma_T)} \right).$$

(24)

By definition then $\gamma_T = \max_{A \subset \mathcal{X}, |A|=T} I(y_{1:T}, f_{1:T}) \geq \frac{1}{2} \ln \left( \frac{\det(K_T + \lambda \Sigma_T)}{\det(\lambda \Sigma_T)} \right)$. On the other hand, $\beta_T$ defined in Lemma 1 can be expanded in a data-independent manner as follows:

$$\beta_T := \sqrt{2 \ln \left( \frac{\det(\lambda \Sigma_T + K_T)^{1/2}}{\delta \det(\lambda \Sigma_T)^{1/2}} \right)} + \sqrt{\lambda} \|f\|_\kappa$$

$$= \sqrt{2 \ln \frac{1}{\delta} + \ln \left( \frac{\det(\lambda \Sigma_T + K_T)}{\det(\lambda \Sigma_T)} \right)} + \sqrt{\lambda} \|f\|_\kappa \leq \sqrt{2 \ln \frac{1}{\delta} + \gamma_T} + \sqrt{\lambda} \mathcal{B}_f. \quad (25)$$

### A.1.3 Bounds for $\gamma_T$

Here, we show the relation between the information gains under heteroscedastic and homoscedastic noise. Note that for the latter the upper bounds are widely known, e.g., [35]. To distinguish between the maximum information gain for heteroscedastic noise with variance-proxy $\rho^2(x)$ and the maximum information gain for homoscedastic noise with fixed variance-proxy $\sigma^2$, we denote them as $\gamma_T^{\rho_x}$ and $\gamma_T^\sigma$ respectively. Recall that $\varrho^2(\cdot) \in [\underline{\varrho}^2, \bar{\varrho}^2]$ for some constant values $\bar{\varrho}^2 \geq \underline{\varrho}^2 > 0$.

Below, we show that $\gamma_T^{\rho_x} \leq \gamma_T^\sigma \frac{\bar{\varrho}^2}{\underline{\varrho}^2}$ with $\sigma^2$ set to $\bar{\varrho}^2$, that only affects the constants but not the main scaling (in terms of $T$) of the known bound for the homoscedastic maximum information gain.

$$\gamma_T^{\rho_x} \overset{①}{=} \max_{A \subset \mathcal{X}, |A|=T} \frac{1}{2} \sum_{t=1}^T \ln \left( 1 + \frac{\sigma_{t-1}^2(x_t|\rho^2(x_t))}{\rho^2(x_t)} \right) \overset{②}{\leq} \max_{A \subset \mathcal{X}, |A|=T} \frac{1}{2} \sum_{t=1}^T \ln \left( 1 + \frac{\sigma_{t-1}^2(x_t|\bar{\varrho}^2)}{\underline{\varrho}^2} \right)$$

(26)

$$\overset{③}{=} \max_{A \subset \mathcal{X}, |A|=T} \frac{1}{2} \sum_{t=1}^T \ln \left( 1 + \frac{\bar{\varrho}^2}{\underline{\varrho}^2} \frac{\sigma_{t-1}^2(x_t|\bar{\varrho}^2)}{\bar{\varrho}^2} \right) \overset{④}{\leq} \max_{A \subset \mathcal{X}, |A|=T} \frac{1}{2} \sum_{t=1}^T \frac{\bar{\varrho}^2}{\underline{\varrho}^2} \ln \left( 1 + \frac{\sigma_{t-1}^2(x_t|\bar{\varrho}^2)}{\bar{\varrho}^2} \right)$$

(27)

$$\overset{⑤}{=} \max_{A \subset \mathcal{X}, |A|=T} \frac{\bar{\varrho}^2}{\underline{\varrho}^2} \frac{1}{2} \sum_{t=1}^T \ln \left( 1 + \frac{\sigma_{t-1}^2(x_t|\sigma^2)}{\sigma^2} \right) = \frac{\bar{\varrho}^2}{\underline{\varrho}^2} \gamma_T^\sigma, \quad (28)$$

where ① follows from Eq. (21). In ②, we lower bound the denominator $\rho^2(x_t)$ and upper bound the numerator $\sigma_{t-1}^2(x_t)$ (due to monotonicity w.r.t. noise variance, i.e., $\sigma_{t-1}^2(x_t|\Sigma_t) \leq \sigma_{t-1}^2(x_t|\bar{\varrho}^2 \mathbf{I}_t)$). In ③, we multiply by $1 = \bar{\varrho}^2/\bar{\varrho}^2$. In ④ we use Bernoulli inequality since $\bar{\varrho}^2/\underline{\varrho}^2 \geq 1$. The obtained expression can be interpreted as a standard information gain for homoscedastic noise and, particularly, with the variance-proxy $\sigma^2$ set to $\bar{\varrho}^2$ due to ⑤. Finally, the upper bounds on $\gamma_T^\sigma$ typically scale sublinearly in $T$ for most of the popularly used kernels [35], e.g, for linear kernel $\gamma_T = \mathcal{O}(d \log T)$, and for squared exponential kernel $\gamma_T = \mathcal{O}(d(\log T)^{d+1})$.

$$\gamma_T = \mathcal{O}(d(\log T)^{d+1})$$

## A.2  Tighter bounds for the variance-proxy $\rho_\eta^2(x)$.

Assumption 2 states that noise $\eta(x)$ from Eq. (13) is $\rho_\eta^2(x)$-sub-Gaussian with variance-proxy $\rho_\eta^2(x)$ being known. In practice, $\rho_\eta^2(x)$ might be unknown. Here, we describe a way to estimate $\rho_\eta^2(x)$ under the following two assumptions: the evaluation noise $\xi(x)$ is strictly sub-Gaussian (that is already reflected in the Assumption 1) and the noise $\eta(x)$ of variance evaluation is also strictly sub-Gaussian, that is, $\mathbb{V}ar[\eta(x)] = \rho_\eta^2(x)$ and $\mathbb{V}ar[\xi(x)] = \rho^2(x)$.

*(i) Reformulation of the sample variance.* We first rewrite the sample variance defined in Eq. (12) as the average over squared differences over all pairs $\{y_1(x), \ldots, y_k(x)\}$:

$$\hat{s}_k^2(x) \overset{\text{\small①}}{=} \frac{1}{2k(k-1)} \sum_{i=1}^{k} \sum_{j=1}^{k} (y_i(x) - y_j(x))^2 \overset{\text{\small②}}{=} \frac{1}{2k(k-1)} \sum_{i=1}^{k} \sum_{j=1}^{k} (\xi_i(x) - \xi_j(x))^2, \quad (29)$$

where ② is due to $y_i(x) = f(x) + \xi_i(x)$, and ① is equivalent to the Eq. (12) due to the following:

$$\frac{1}{2k(k-1)} \sum_{i=1}^{k} \sum_{j=1}^{k} (y_i - y_j)^2 = \frac{1}{2k(k-1)} \sum_{i=1}^{k} \sum_{j=1}^{k} (y_i - \hat{m}_k + \hat{m}_k - y_j)^2 \quad (30)$$

$$= \frac{1}{2k(k-1)} \sum_{i=1}^{k} \sum_{j=1}^{k} \left[ (y_i - \hat{m}_k)^2 + (y_j - \hat{m}_k)^2 - 2(y_i - \hat{m}_k)(y_j - \hat{m}_k) \right]$$

$$= \frac{1}{2k(k-1)} \sum_{i=1}^{k} \sum_{j=1}^{k} \left[ (y_i - \hat{m}_k)^2 + (y_j - \hat{m}_k)^2 \right]$$

$$= \frac{1}{k-1} \sum_{i=1}^{k} (y_i - \hat{m}_k)^2. \quad (31)$$

*(ii) Variance of the sample variance* $\mathbb{V}ar[\hat{s}_k^2(x)]$. In Eq. (29), we show that sample variance can be written in terms of the noise $\xi(x)$. In [4] (see Eq. (37)), it is shown that for i.i.d observations $\{\xi_1(x), \ldots, \xi_k(x)\}$, sampled from a distribution with the 2nd and 4th central moments $\mathbb{V}ar[\xi(x)]$ and $\mu_4(x) = \mathbb{E}[\xi^4(x)]$, respectively, the variance of the sample variance can be computed as follows:

$$\mathbb{V}ar[\hat{s}_k^2(x)] = \mathbb{E}[(\hat{s}_k^2(x))^2] - \mathbb{E}[\hat{s}_k^2(x)]^2 = \frac{\mu_4(x)}{k} - \frac{(k-3)\mathbb{V}ar^2[\xi(x)]}{k(k-1)}.$$

Since $\xi(x)$ is strictly $\rho(x)$–sub-Gaussian, the latter can be further adapted as

$$\mathbb{V}ar[\hat{s}_k^2(x)] = \frac{\mu_4(x)}{k} - \frac{(k-3)\rho^4(x)}{k(k-1)}.$$

*(iii)* Due to $\eta(x)$ being strictly sub-Gaussian, i.e., $\rho_\eta^2(x) = \mathbb{V}ar[\eta(x)] = \mathbb{V}ar[\hat{s}_k^2(x)]$, the derivation above also holds for the variance-proxy $\rho_\eta^2(x)$:

$$\rho_\eta^2(x) = \frac{\mu_4(x)}{k} - \frac{(k-3)\rho^4(x)}{k(k-1)}.$$

*(iv) Bound 4th moment* $\mu_4(x)$. The 4th moment $\mu_4(x)$ can expressed in terms of the distribution kurtosis that is bounded under our assumptions. Particularly, *kurtosis* $\mathrm{Kurt}[\xi] := \frac{\mathbb{E}[(\xi-\mathbb{E}[\xi])^4]}{\mathbb{V}ar^2(\xi)}$ is measure that identifies the tails behaviour of the distribution of $\xi$; $\mathrm{Kurt}(\xi) = 3$ for normally distribute $\xi$ and $\mathrm{Kurt}(\xi) \leq 3$ for strictly sub-Gaussian random variable $\xi$ (see [2]). This implies

$$\mu_4(x) = \mathrm{Kurt}(\xi(x))\rho^4(x) \leq 3\rho^4(x).$$

*(v) Bound variance-proxy.* There

$$\rho_\eta^2(x) \leq \frac{3(k-1)\rho^4(x) - (k-3)\rho^4(x)}{k(k-1)} = \frac{3k-3-k+3}{k(k-1)}\rho^4(x) = \frac{2\rho^4(x)}{k-1}.$$

In case of the known bound $\bar{\varrho}^2 \geq \rho^2(x)$, we bound the unknown $\rho_\eta^2(x)$ as follows:

$$\rho_\eta^2(x) \leq \frac{2\bar{\varrho}^4}{k-1}.$$

## A.3  Method details: GP-estimator of variance-proxy $\rho^2$

According to the Assumption 2, variance-proxy $\rho^2 \in \mathcal{H}_{\kappa^{var}}$ is smooth, and $\eta(x) = \hat{s}_k^2(x) - \rho^2(x)$ is $\rho_\eta(x)$-sub-Gaussian with known variance-proxy $\rho_\eta^2(x)$. In this case, confidence bounds for $\rho^2(x)$ follow the ones derived in Lemma 1 with $\beta_t^{var}$ based on $\Sigma_t^{var}$. Particularly, we collect noise variance evaluations $\{x_t, \hat{s}_k(x_t)\}_{t=0}^T$. Then the estimates for $\mu_T^{var}(x)$ and $\sigma_T^{var}(x)$ for $\rho^2$ follow the corresponding estimates for $f(x)$. Particularly,

$$\mu_t^{var}(x) = \kappa_t^{var}(x)^T(K_t^{var} + \lambda\Sigma_t^{var})^{-1}\hat{s}_{1:t}, \tag{32}$$

$$\sigma_t^{var}(x)^2 = \frac{1}{\lambda}(\kappa^{var}(x,x) - \kappa_t^{var}(x)^\top(K_t^{var} + \lambda\Sigma_t^{var})^{-1}\kappa_t^{var}(x)), \tag{33}$$

where $\Sigma_t^{var} = \mathrm{diag}[\rho_\eta^2(x_1), \ldots, \rho_\eta^2(x_t)]$, $\kappa_t^{var}(x) = [\kappa^{var}(x_1, x), \ldots, \kappa^{var}(x_t, x)]^T$ and $(K_t^{var})_{i,j} = \kappa^{var}(x_i, x_j)$. The confidence bounds are then:

$$\mathrm{ucb}_t^{var}(x) = \mu_{t-1}^{var}(x) + \beta_t^{var}\sigma_{t-1}^{var}(x)$$
$$\mathrm{lcb}_t^{var}(x) = \mu_{t-1}^{var}(x) - \beta_t^{var}\sigma_{t-1}^{var}(x),$$

with $\{\beta_t^{var}\}_{t=1}^T$ set according to Lemma 1.

## A.4  Proof of Theorem 1

**Theorem 1.** *Consider any $f \in \mathcal{H}_\kappa$ with $\|f\|_\kappa \leq \mathcal{B}_f$ and sampling model in Eq. (1) with unknown variance-proxy $\rho^2(x)$ that satisfies Assumptions 1 and 2. Let $\{x_t\}_{t=1}^T$ denote the set of actions chosen by* RAHBO *(Algorithm 1) over T rounds. Set $\{\beta_t\}_{t=1}^T$ and $\{\beta_t^{var}\}_{t=1}^T$ according to Lemma 1 with $\lambda = 1$, $\mathcal{R}^2 = \max_{x \in \mathcal{X}} \rho_\eta^2(x_t)$ and $\rho(\cdot) \in [\underline{\varrho}, \bar{\varrho}]$. Then, the risk-averse cumulative regret $R_T$ of* RAHBO *is bounded as follows:*

$$\Pr\left\{R_T \leq \beta_T k\sqrt{\frac{2T\hat{\gamma}_T}{\ln(1 + k/\bar{\varrho}^2)}} + \alpha\beta_T^{var}k\sqrt{\frac{2T\Gamma_T}{\ln(1 + \mathcal{R}^{-2})}}, \quad \forall T \geq 1\right\} \geq 1 - \delta. \tag{34}$$

*Proof.* The main steps of our proof are as follows: In *Step 1*, we derive the upper and the lower confidence bounds, $\mathrm{ucb}_t^{\mathrm{MV}}(x_t)$ and $\mathrm{lcb}_t^{\mathrm{MV}}(x_t)$, on $\mathrm{MV}(x_t)$ at iteration $t$. In *Step 2*, we bound the instantaneous risk-averse regret $r(x_t) := \mathrm{MV}(x^*) - \mathrm{MV}(x_t)$. In *Step 3*, we derive mutual information both for function and variance-proxy evaluations. In *Step 4*, we bound the sum of variances via mutual information. In *Step 5*, we bound the cumulative regret $R_T = \sum_{t=1}^T r(x_t)$ based on the previous steps.

***Step 1: On confidence bounds for MV(x).***

*(i) On confidence bounds for $\rho^2(x)$.* According to Eq. (33), with probability $1 - \delta$ the following confidence bounds hold with $\{\beta_t^{var}\}_{t=1}^T$ set according to Lemma 1:

$$\mathrm{ucb}_t^{var}(x) = \mu_{t-1}^{var}(x) + \beta_t^{var}\sigma_{t-1}^{var}(x),$$
$$\mathrm{lcb}_t^{var}(x) = \mu_{t-1}^{var}(x) - \beta_t^{var}\sigma_{t-1}^{var}(x).$$

*(ii) On confidence bounds for $f(x)$.* Here we adapt confidence bounds introduced in Eq. (18)-(19) since Eq. (5) relies on the unknown variance-proxy $\rho^2(x)$ incorporated into $\Sigma_T$. Conditioning on the event that $\rho^2(x)$ is upper bounded by $\mathrm{ucb}_t^{var}(x) \geq \rho(x)^2$ defined in (i), the confidence bounds for $f$ with probability $1 - \delta$ are:

$$\mathrm{ucb}_t^f(x) = \mu_{t-1}(x|\hat{\Sigma}_{t-1}) + \beta_t\sigma_{t-1}(x|\hat{\Sigma}_{t-1}), \tag{35}$$

$$\mathrm{lcb}_t^f(x) = \mu_{t-1}(x|\hat{\Sigma}_{t-1}) - \beta_t\sigma_{t-1}(x|\hat{\Sigma}_{t-1}), \forall x, t. \tag{36}$$

*(iii) On confidence bounds for MV(x).* Finally, combining (i) and (ii) and using the union bound, with probability $1 - 2\delta$, we get $\text{lcb}_t^{\text{MV}}(x) \le \text{MV}(x) \le \text{ucb}_t^{\text{MV}}(x)$ with

$$\text{ucb}_t^{\text{MV}}(x) = \text{ucb}_t^f(x) - \alpha\text{lcb}_t^{var}(x), \tag{37}$$

$$\text{lcb}_t^{\text{MV}}(x) = \text{lcb}_t^f(x) - \alpha\text{ucb}_t^{var}(x), \ \forall x, t. \tag{38}$$

***Step 2: On bounding the instantaneous regret.***
First, we bound instantaneous regret of a single measurement at point $x_t$, but with unknown variance-proxy $\rho^2(x)$ as follows:

$$\begin{aligned}
r_t := \text{MV}(x^*) - \text{MV}(x_t) &\le \text{ucb}_t^{\text{MV}}(x^*) - \text{lcb}_t^{\text{MV}}(x_t) \\
&\le \text{ucb}_t^{\text{MV}}(x_t) - \text{lcb}_t^{\text{MV}}(x_t) \\
&= \text{ucb}_t^f(x_t) - \text{lcb}_t^f(x_t) + \alpha(\text{ucb}_t^{var}(x_t) - \text{lcb}_t^{var}(x_t)) \\
&= 2\beta_t\sigma_{t-1}(x_t|\hat{\Sigma}_{t-1}) + 2\alpha\beta_t^{var}\sigma_{t-1}^{var}(x_t). \tag{39}
\end{aligned}$$

The second inequality is due to the acquisition algorithm. The last equality is due to the fact that $\text{ucb}_t^f(x) - \text{lcb}_t^f(x) = 2\beta_t\sigma_{t-1}(x_t)$ by definition, as well as $\text{ucb}_t^{var}(x) - \text{lcb}_t^{var}(x) = 2\beta_t^{var}\sigma_{t-1}^{var}(x_t)$.

Note that at each iteration $t$ we take $k$ measurements, hence the total number of measurements is $Tk$. Thus, we can bound the cumulative regret by

$$\begin{aligned}
R_T = \sum_{t=1}^{T} kr(x_t) &\le k\sum_{t=1}^{T} 2\beta_t\sigma_{t-1}(x_t|\hat{\Sigma}_{t-1}) + k\sum_{t=1}^{T} 2\alpha\beta_t^{var}\sigma_{t-1}^{var}(x_t) \\
&\le 2k\beta_T\sum_{t=1}^{T} \sigma_{t-1}(x_t|\hat{\Sigma}_{t-1}) + 2k\alpha\beta_T^{var}\sum_{t=1}^{T} \sigma_{t-1}^{var}(x_t). \tag{40}
\end{aligned}$$

***Step 3: On bounding maximum information gain.***
We follow the notion of information gain $I(\hat{m}_{1:T}, f_{1:T})$ computed assuming that $\hat{m}_{1:T} = [\hat{m}_k(x_1), \ldots, \hat{m}_k(x_T)]^T$ with $\hat{m}_k(x_t) = \frac{1}{k}\sum_{i=1}^{k} y_i(x_t)$ (Eq. (12)). Under the modelling assumptions $f_{1:T} \sim \mathcal{N}(0, \lambda^{-1}K_T)$, and $\hat{m}_{1:T} \sim \mathcal{N}(f_{1:T}, \text{diag}(\bar{\varrho}^2/k))$ with variance-proxy $\bar{\varrho}^2/k$, the information gain is:

$$I(\hat{m}_{1:T}, f_{1:T}) := \sum_{t=1}^{T} \frac{1}{2}\ln\left(1 + \frac{\sigma_{t-1}^2(x_t|\text{diag}(\bar{\varrho}^2/k))}{\bar{\varrho}^2/k}\right). \tag{41}$$

We define the corresponding maximum information gain $\hat{\gamma}_T = \max_{A \subset \mathcal{X}, |A|=T} I(\hat{m}_{1:T}, f_{1:T})$

$$\hat{\gamma}_T := \max_{A \subset \mathcal{X}, |A|=T} \sum_{t=1}^{T} \frac{1}{2}\ln\left(1 + \frac{\sigma_{t-1}^2(x_t|\text{diag}(\bar{\varrho}^2/k))}{\bar{\varrho}^2/k}\right). \tag{42}$$

Analogously, for $\rho(x)$ with the posterior $\mathcal{N}(\mu_t^{var}(x), (\sigma_t^{var}(x))^2)$, the information gain is defined as:

$$I(\hat{s}_{1:T}^2, \rho_{1:T}^2) := \frac{1}{2}\sum_{t=1}^{T} \ln\left(1 + \frac{(\sigma_{t-1}^{var})^2(x)}{\rho_\eta^2(x_t)}\right). \tag{43}$$

Then, the corresponding maximum information gain $\Gamma_T$ is as follows:

$$\Gamma_T := \max_{A \subset \mathcal{X}, |A|=T} I(\hat{s}_{1:T}^2, \rho_{1:T}^2) = \max_{A \subset \mathcal{X}, |A|=T} \frac{1}{2}\sum_{t=1}^{T} \ln\left(1 + \frac{(\sigma_{t-1}^{var})^2(x)}{\rho_\eta^2(x_t)}\right), \tag{44}$$

where $A$ is again a set of size $T$ with points $\{x_1, \ldots, x_T\}$.

**Step 4**: *On bounding* $\sum_{t=1}^{T} \sigma_{t-1}(x_t|\hat{\Sigma}_{t-1})$ *and* $\sum_{t=1}^{T} \sigma_{t-1}^{var}(x_t)$

We repeat the corresponding derivation for known $\rho^2(x)$, recalling that $\rho^2(x) \le \bar{\varrho}^2, \forall x \in \mathcal{X}$:

$$\sum_{t=1}^{T} \sigma_{t-1}(x_t|\hat{\Sigma}_{t-1}) = \sum_{t=1}^{T} \frac{\bar{\varrho}}{\bar{\varrho}} \sigma_{t-1}(x_t|\hat{\Sigma}_{t-1}) \le \sqrt{T \sum_{t=1}^{T} \frac{\bar{\varrho}^2}{k} \frac{\sigma_{t-1}^2\big(x_t|\mathrm{diag}(\bar{\varrho}^2/k)\big)}{\bar{\varrho}^2/k}}$$

$$\le \sqrt{T \frac{\bar{\varrho}^2}{k} \frac{k/\bar{\varrho}^2}{\ln(1+k/\bar{\varrho}^2)} \sum_{t=1}^{T} \ln\left(1 + \frac{\sigma_{t-1}^2\big(x_t|\mathrm{diag}(\bar{\varrho}^2/k)\big)}{\bar{\varrho}^2/k}\right)}$$

$$\le \sqrt{\frac{2T}{\ln(1+k/\bar{\varrho}^2)} \underbrace{\sum_{t=1}^{T} \frac{1}{2} \ln\left(1 + \frac{\sigma_{t-1}^2\big(x_t|\mathrm{diag}(\bar{\varrho}^2/k)\big)}{\bar{\varrho}^2/k}\right)}_{\text{mutual information Eq. (41)}}}. \tag{45}$$

Here, the first inequality follows from Cauchy-Schwarz inequality and the fact that $\sigma_t(x_t|\hat{\Sigma}_t) \le \sigma_t(x_t|\mathrm{diag}(\bar{\varrho}^2/k))$. The latter holds by the definition of $\hat{\Sigma}_t$, particularly:

$$\sigma_t^2(x_t|\hat{\Sigma}_t) = \frac{1}{\lambda}(\kappa(x,x) - \kappa_t(x)^\top (K_t + \lambda\hat{\Sigma}_t)^{-1} \kappa_t(x)),$$

$$\sigma_t^2\big(x_t|\mathrm{diag}(\bar{\varrho}^2/k)\big) = \frac{1}{\lambda}(\kappa(x,x) - \kappa_t(x)^\top \big(K_t + \lambda\mathrm{diag}(\bar{\varrho}^2/k)\big)^{-1} \kappa_t(x)),$$

$$\hat{\Sigma}_t = \tfrac{1}{k}\mathrm{diag}\big(\min\{\mathrm{ucb}_t^{var}(x_1), \bar{\varrho}^2\}, \ldots, \min\{\mathrm{ucb}_t^{var}(x_t), \bar{\varrho}^2\}\big),$$

then $\hat{\Sigma}_t \preceq \mathrm{diag}(\bar{\varrho}^2/k)$, and $-(K_t + \lambda\hat{\Sigma}_t)^{-1} \preceq -(K_t + \lambda\mathrm{diag}(\bar{\varrho}^2/k))^{-1}$. That implies

$$\sigma_t^2(x_t|\hat{\Sigma}_t) - \sigma_t^2\big(x_t|\mathrm{diag}(\bar{\varrho}^2/k)\big) = -\kappa_t(x)^\top \big(K_t + \lambda\hat{\Sigma}_t\big)^{-1} \kappa_t(x)$$
$$+ \kappa_t(x)^\top (K_t + \lambda\mathrm{diag}(\bar{\varrho}^2/k))^{-1} \kappa_t(x)$$
$$\le 0.$$

The second inequality in Eq. (45) is due to the fact that for any $s^2 \in [0, k/\bar{\varrho}^2(x_t)]$ we can bound $s^2 \le \frac{k/\bar{\varrho}^2(x_t)}{\ln(1+k/\bar{\varrho}^2(x_t))} \ln(1+s^2)$, that also holds for $s^2 := \frac{\sigma_{t-1}^2\big(x_t|\mathrm{diag}(\bar{\varrho}^2/k)\big)}{\bar{\varrho}^2/k}$ since for any $\lambda \ge 1$

$$\frac{\sigma_{t-1}^2\big(x_t|\mathrm{diag}(\bar{\varrho}^2/k)\big)}{\bar{\varrho}^2/k} \le \frac{\lambda^{-1}\kappa(x_t,x_t)}{\bar{\varrho}^2/k} \le k/\bar{\varrho}^2.$$

Similarly, we bound

$$\sum_{t=1}^{T} \sigma_{t-1}^{var}(x_t) = \sum_{t=1}^{T} \frac{\rho_\eta(x_t)}{\rho_\eta(x_t)} \sigma_{t-1}^{var}(x_t) \le \sqrt{T \sum_{t=1}^{T} \rho_\eta^2(x_t) \frac{(\sigma_{t-1}^{var})^2(x_t)}{\rho_\eta^2(x_t)}}$$

$$\le \sqrt{\frac{2T}{\ln(1+\mathcal{R}^{-2})} \underbrace{\sum_{t=1}^{T} \frac{1}{2} \ln\left(1 + \frac{(\sigma_{t-1}^{var})^2(x_t)}{\rho_\eta^2(x_t)}\right)}_{\text{mututal information Eq. (43)}}}, \tag{46}$$

in the above we define $\mathcal{R}^2 := \max_{x \in A} \rho_\eta^2(x), A = \{x_1, \ldots, x_T\}$.

**Step 5**: *On bounding cumulative regret* $R_T = \sum_{t=1}^{T} kr(x_t)$

Combining the above three steps together, we obtain with probability $1 - 2\delta$

$$R_T \le \beta_T k \sqrt{\frac{2T\hat{\gamma}_T}{\ln(1+k/\bar{\varrho}^2)}} + \alpha\beta_T^{var} k \sqrt{\frac{2T\Gamma_T}{\ln(1+\mathcal{R}^{-2})}}. \tag{47}$$

## A.5 Proof of Corollary 1.1

**Corollary 1.1** *Consider the setup of Theorem 1. Let $A = \{x_t\}_{t=1}^T$ denote actions selected by RAHBO over $T$ rounds. Then, with probability at least $1 - \delta$, the reported point $\hat{x}_T := \arg\max_{x_t \in A} \mathrm{lcb}_t^{MV}(x_t)$, where $\mathrm{lcb}_t^{MV}(x_t) = \mathrm{lcb}_t^f(x) - \alpha\, \mathrm{ucb}_t^{var}(x)$, achieves $\epsilon$-accuracy, i.e., $MV(x^*) - MV(\hat{x}_T) \leq \epsilon$, after $T \geq \frac{4\beta_T^2 \hat{\gamma}_T / \ln(1 + k/\bar{\varrho}^2) + 4\alpha(\beta_t^{var})^2 \Gamma_T / \ln(1 + \mathcal{R}^{-2})}{\epsilon^2}$ rounds.*

*Proof.* We select the maximizer of $\mathrm{lcb}_t^{MV}(x_t)$ over the past points $x_t$:

$$\hat{x}_T := x_{t^*}, \text{ where } t^* := \arg\max_t \{\mathrm{lcb}_t^{MV}(x_t)\} = \arg\min_t\{MV(x^*) - \mathrm{lcb}_t^{MV}(x_t)\},$$

since adding a constant does not change the solution. We denote $\hat{r}(x_t) := MV(x^*) - \mathrm{lcb}_t^{MV}(x_t)$. Then we obtain the following bound

$$MV(x^*) - MV(x_{t^*}) \leq MV(x^*) - \mathrm{lcb}_{t^*}^{MV}(x_{t^*}) = \frac{1}{T}\sum_{t=1}^T \hat{r}(x_{t^*})$$

$$\leq \frac{1}{T}\sum_{t=1}^T \hat{r}(x_t) = \frac{1}{T}\sum_{t=1}^T \left(MV(x^*) - \mathrm{lcb}_t^{MV}(x_t)\right)$$

$$\leq \frac{1}{T}\sum_{t=1}^T \left(\mathrm{ucb}_t^{MV}(x^*) - \mathrm{lcb}_t^{MV}(x_t)\right)$$

$$\leq \frac{1}{T}\sum_{t=1}^T \left(\mathrm{ucb}_t^{MV}(x_t) - \mathrm{lcb}_t^{MV}(x_t)\right). \tag{48}$$

In the above, the first inequality holds with high probability by definition $\mathrm{lcb}_{t^*}^{MV}(x_{t^*}) \leq MV(x_{t^*})$, the second inequality is due to $t^* := \arg\min_t \hat{r}(x_t)$ and therefore $\hat{r}(x_{t^*}) \leq \hat{r}(x_t)\ \forall t = 1, \ldots, T$. The third inequality holds since $\mathrm{ucb}_t^{MV}(x) \geq MV(x)$ with high probability, and the fourth is due to $\mathrm{ucb}_t^{MV}(x_t) \geq \mathrm{ucb}_t^{MV}(x)$ for every $x$, since $x_t$ is selected via Algorithm 1.

Recalling Eq. (47), note that the following bounds hold:

$$R_T = \sum_{t=1}^T kr(x_t) \leq \sum_{t=1}^T k(\mathrm{ucb}_t^{MV}(x_t) - \mathrm{lcb}_t^{MV}(x_t)) \leq \beta_T k\sqrt{\frac{2T\hat{\gamma}_T}{\ln(1 + k/\bar{\varrho}^2)}} + \alpha\beta_T^{var}k\sqrt{\frac{2T\Gamma_T}{\ln(1 + \mathcal{R}^{-2})}}. \tag{49}$$

Combining the above Eq. (49) with Eq. (48) we can get the following upper bound

$$MV(x^*) - MV(x_{t^*}) \leq \frac{\beta_T k\sqrt{2T\hat{\gamma}_T/\ln(1 + k/\bar{\varrho}^2)} + \alpha\beta_T^{var}k\sqrt{2T\Gamma_T/\ln(1 + \mathcal{R}^{-2})}}{kT}$$

$$\leq \frac{\sqrt{4\left(k\beta_T^2\hat{\gamma}_T/\ln(1 + k/\bar{\varrho}^2) + \alpha k(\beta_t^{var})^2\Gamma_T/\ln(1 + \mathcal{R}^{-2})\right)}}{\sqrt{kT}}.$$

Therefore, for $Tk$ samples with $Tk \geq \frac{4(k\beta_T^2\hat{\gamma}_T/\ln(1 + k/\bar{\varrho}^2) + \alpha k(\beta_t^{var})^2\Gamma_T/\ln(1 + \mathcal{R}^{-2}))}{\epsilon^2}$ we finally obtain

$$MV(x^*) - MV(x_{t^*}) \leq \epsilon.$$

$\square$

## A.6 Experimental settings and extended results

**Implementation and resources** We implemented all our experiments using Python and BoTorch [3].[3] We ran our experiments on an Intel(R) Xeon(R) CPU E5-2699 v3 @ 2.30GHz machine.

---

[3]`https://botorch.org/`

### A.6.1 Example function

We provide additional visualizations for the example sine function in Fig. 6. These examples demonstrate that exploration-exploitation trade-off (as in GP-UCB) might not be enough to prefer points with lower noise and GP-UCB might tend to acquire points with higher variance. In contrast, RAHBO, initialized with the same point, prefers points with lower risk inherited in noise.

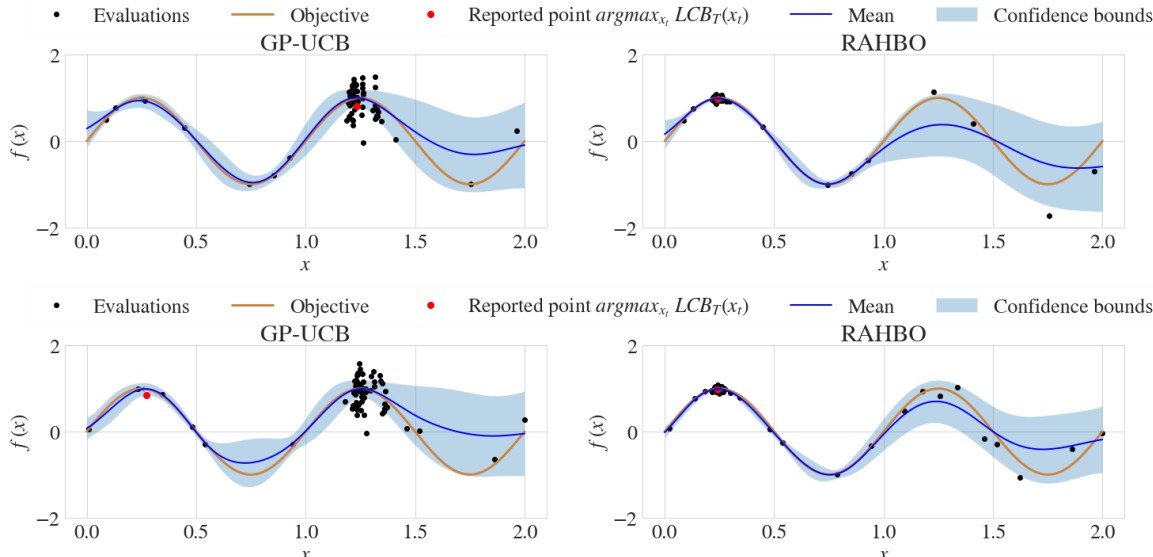

Figure 6: Additional examples for Section 4 (each row corresponds to one initialization). GP models fitted for GP-UCB (left column) and RAHBO (right column) for sine function. After initialization with the same sampled points, GP-UCB concentrates on the high-noise region whereas RAHBO prefers small variance.

### A.6.2 Branin

We provide additional visualizations, experimental details and results. Firstly, we plot the noise-perturbed objective function in Fig. 7 in addition to the visualization in Fig. 1c. In Fig. 8, we plot cumulative regret and simple mean-variance regrets that extends the results in Fig. 5a with RAHBO-US. The general setting is the same as described for Fig. 5a: we use 10 initial samples, repeat each evaluation $k = 10$ times, and RAHBO-US additionally uses 10 samples for learning the variance function with uncertainty sampling. During the optimization, RAHBO-US updates the GP model for variance function after every acquired point.

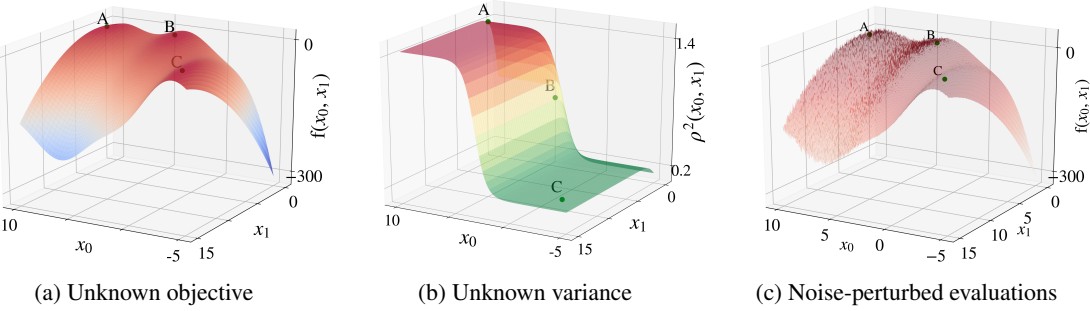

(a) Unknown objective  (b) Unknown variance  (c) Noise-perturbed evaluations

Figure 7: Visualization of noise-perturbed function landscape:(a) Unknown objective with 3 global maxima marked as (A, B, C). (b) Heteroscedastic noise variance over the same domain: the noise level at (A, B, C) varies according to the sigmoid function. (c) Noise-perturbed evaluations: A is located in the noisiest region.

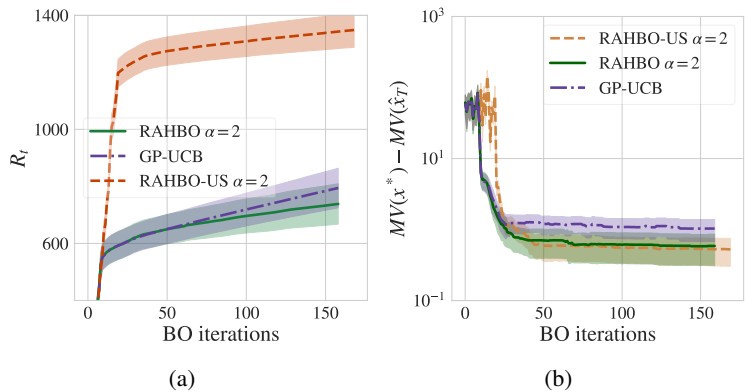

<p align="center">(a)                              (b)</p>

Figure 8: **Branin:** (a) Cumulative regret. (b) Suboptimality w.r.t. MV

### A.7 Random Forest tuning

**Experiment motivation:** Consider the motivating example first: the optimized RF model will be exploited under the data drift over time, e.g., detecting fraud during a week. We are interested not only in high performance *on average* but also in low variance across the results. Particularly, the first can be a realization of the decent result in the first days and unacceptable result in the last days, and the latter ensures lower dispersion over the days while keeping a reasonable mean. In this case, when training an over-parametrized model that is prone to overfitting (to the training data), e.g., Random Forest (RF) with deep trees, high variance in validation error might be observed. In contrast, a model that is less prone to overfitting can result into a similar validation error with lower variance.

**RF specifications:** We use scikit-learn implementation of RF. The RF search spaces for BO are listed in Table 1 and other parameters are the default provided by scikit-learn. [4] During BO, we transform the parameter space to the unit-cube space.

**Dataset:** We tune RF on a dataset of fraudulent credit card transactions [23] originally announced for Kaggle competition.[5] It is a highly imbalanced dataset that consists of 285k transactions and only 0.2% are fraud examples. The transactions occurred in two days and each has a time feature that contains the seconds elapsed between each transaction and the first transaction in the dataset. We use the time feature to split the data into train and validation sets such that validation transactions happen later than the training ones. The distribution of the fraud and non-fraud transactions in time is presented in Fig. 9.

In BO, we collect evaluation in the following way: we fix the training data to be the first half of the transactions, and the rest we split into 5 validation folds that are consecutive in time. The RF model is then trained on the fixed training set, and evaluated on the validations sets. We use a balanced accuracy score that takes imbalance in the data into account.

| task | hyperparameter | search space |
|---|---|---|
| | n_estimators | [1, 100] |
| RandomForest | max_features | [5, 28] |
| | max_depth | [1, 15] |

Table 1: Search space description for RF.

---

[4] https://scikit-learn.org/stable/modules/generated/sklearn.ensemble. RandomForestClassifier.html

[5] https://www.kaggle.com/mlg-ulb/creditcardfraud

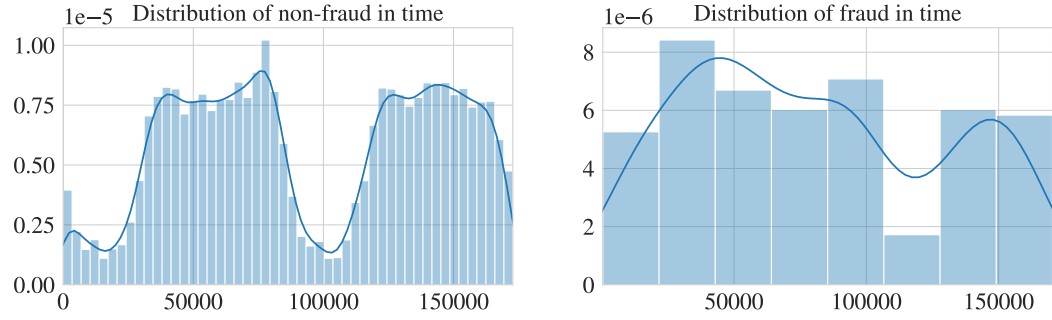

Figure 9: Distribution of non-fraud (left) and fraud (right) transactions in the dataset

### A.7.1 Tuning Swiss free-electron laser (SwissFEL)

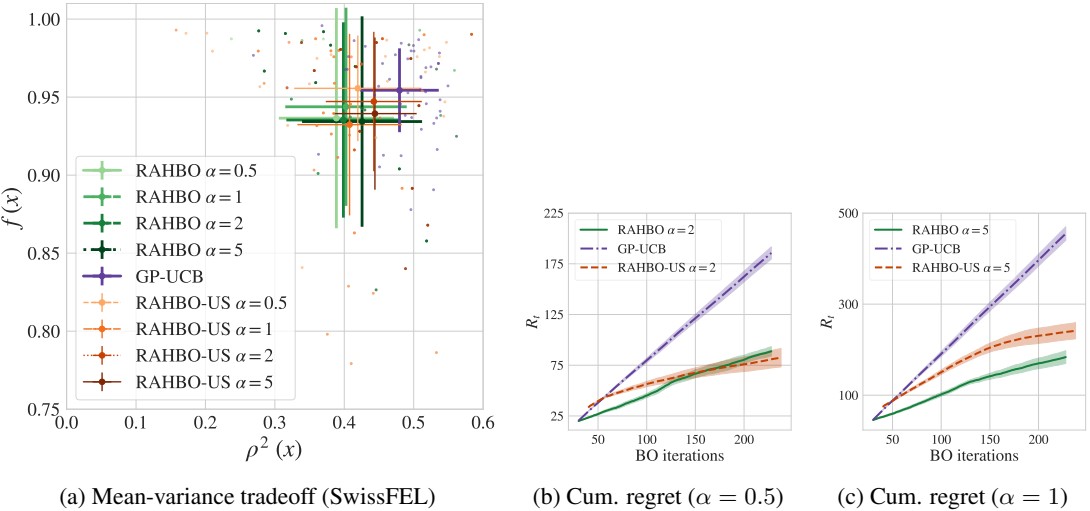

(a) Mean-variance tradeoff (SwissFEL)      (b) Cum. regret ($\alpha = 0.5$)      (c) Cum. regret ($\alpha = 1$)

Figure 10: (a) We plot standard deviation error bars for $f(x)$ and $\rho^2(x)$ at the *reported point* by the best observed value $x^{(T)} = \arg\max_{x_t} y_t(x_t)$ after BO completion for SwissFEL. The mean and std of the error bars are taken over the repeated BO experiments. The results demonstrate, that reporting based on the best observed value inherits high noise and as the result all methods perform similarly. Intuitively, when noise variance is high, it is possible to observe higher values. That however also inherits observing much lower value at this point, this leading to very non-robust solutions. (b-c) Cumulative regret for $\alpha = 2$ and $\alpha = 5$.