# OpenReview forum: "Risk-averse Heteroscedastic Bayesian Optimization"
_NeurIPS.cc/2021/Conference — NeurIPS 2021 Poster_

### Official Review · Reviewer_hxC9 · 2021-07-15

**Rating:** 7
**Confidence:** 4

**Summary:**

This paper presents an approach for risk-averse Bayesian Optimization in the case that the function to be optimized is subject to heteroskedastic observation noise. For a mean-variance model, the authors propose an acquisition function and provide bound on the cumulative regret for the case when the variance proxy is known. They then extend this to model the unknown variance proxy (assuming strict sub-gaussianity) and provide regret bounds also for this case. The algorithm is empirically evaluated on a set of synthetic and real problems, where it is shown to improve upon non-robust BO approaches in terms of the regret defined.

**Ethical Concerns:**

No ethical concerns

**Limitations And Societal Impact:**

- In my mind, the main practical limitation of the paper is that the parameter alpha will often be unknown in practice (w/o already having a reasonably good understanding of both the behavior of the function and the noise). The authors have not really addressed this and should expand on it in their discussion.
- The authors state that "In settings where the noise is homoscedastic, our approach does not bring additional value". I think there is more to that, likely sample efficiency will suffer compared to approaches that were not designed with robustness in mind. Ideally this could be quantified in an ablation study, and at the very least discussed.

**Main Review:**

### High-level summary:
- The problem setting is relevant of interest for the community.
- The discussion of related work is overall good.
- The contributions are novel and appear technically sound.
- The paper is generally well written and easy to follow.
- The empirical evaluation is solid, but could benefit from additional baselines.
- Overall a good paper; with room for improvement, but a pretty clear accept in my mind.


### Notes:
- Why are all of the named applications "high-stakes" applications? This is understandable for drug discovery and robotics, but not really for the other examples, please substantiate.
  - More generally, the paper could do a better job at explaining the practical implications of this work by giving concrete examples.
- It is quite a bit into the paper before the authors discuss that the setting of heteroskedastic noise and the setting of risk-averse BO are not new but have been studied before. They should do  this in the introduction to avoid any potential misconceptions and clearly identify the novelty of their work over existing literature.
- The discussion of related work is relatively comprehensive. However, the setting of this paper is quite close to that of [1] (which focuses purely on modeling). The authors should add that reference and discuss how their work relates to that of [1].
- I am not fully convinced that the mean-variance-model is a practical choice; it seems much harder for me to properly choose the coefficient of absolute risk tolerance than, say, to optimizat VaR or CVaR. From my experience working with practitioners, the assumption that alpha is fixed and known will generally not hold in practice. It would be good to address this rather than stating it as an assumption without any discussion.
- The formulation of the acquisition function (eq 9) is pretty intuitive and the straightforward choice. An alternative would be to not consider the posterior variance sigma(x)  of the latent function, but consider the the overall posterior predictive sqrt(sigma(x)^2 + rho(x)^2). It would be worth discussing such option and compare the choice made here to other approaches.
- The paper could benefit from additional discussion of the algorithm. For instance, in eq. 9 it may appear that there is a tension between beta and alpha, and it took me a bit to think through the difference implication of the posterior uncertainty of the latent function and the variance of the observation. It would be helpful to provide this kind of intuition for what is going on to the reader.
- The repeated experiment setting is a natural thing to do but appears limiting, and it seems like it could really hurt sample efficiency. With a smoothness assumption on the behavior of the variance proxy, it should be possible to devise a more sample efficient algorithm than simply re-evaluating each candidate k times. This is discussed also in [1] and the authors should discuss how their approach compares to that of [1].
- The theoretical results appear sound, though I did not check them in any detail. It's nice to show some regret bounds, and this appears to require a good amount of technical finesse. However, I feel like value of these theoretical contributions for practical purposes is somewhat limited. For instance, the results rely on finding the maximizer of the acquisition function exactly, but in actual implementations this requires solving very hard non-convex optimization problems repeatedly, where and one generally cannot guarantee finding the globally optimal solution for all but the smallest problems.
  - The theoretical results do get quite technical, it would be useful for the non-expert to have some high-level summary for how the theoretical analysis and proof technique relates to that in Srinivas et al.
- The empirical results are quite convincing and are presented well. What I feel is missing are some other existing robust BO methods (e.g. optimizing VaR or CVaR). While those may not specifically target the mean-variance model formulation, they could still perform quite well in terms of the regret defined here. The method from [3] has been implemented in botorch and should thus be quite easy to hook up into the submitted code: https://github.com/pytorch/botorch/blob/master/tutorials/risk_averse_bo_with_environmental_variables.ipynb
- The FEL setup sounds quite interesting, it would be nice to get some more details into the non-anonymized version.
- The authors state that "In settings where the noise is homoscedastic, our approach does not bring additional value". I think there is more to that, likely sample efficiency will suffer compared to approaches that were not designed with robustness in mind. It could be valuable to provide some analysis of the performance in case the noise is indeed homoskedastic; this would allow to get some idea of the "price of being robust to heteroskedasticity". To understand this, it would be good to have an ablation study that uses a homoskedasatic setting.
- Please honor botorch's request for citation https://github.com/pytorch/botorch#citing-botorch [2]
- Misc:
    - l. 229: be clear what you mean by heteroskedastic GP (looks like it's a botorch FixedNoiseGP)
    - Fig 5: "Simple regret fat the reprted"
    - The term "zeroth-order observation" may not be clear to everyone, consider rephrasing (no gradient info)


### Questions:
- Can you explain the difference between your modeling setup and the HeteroskedasticSingleTaskGP from https://github.com/pytorch/botorch/blob/master/botorch/models/gp_regression.py#L339 ?
- The fact that `k` (the number of function samples to estimate the local variance) is fixed and non-adaptive is somewhat disappointing. It should be possible to improve sample efficiency by in later stages making this adaptive to the posterior uncertainty of the variance proxy at the point to be evaluated. In fact, you could be Bayesian about this, too. This would mean interpreting the posterior distribution of the variance proxy as your prior, and then the `k` samples as observations, and then use the observations to update this prior. Did you consider this?

### References:

[1] K. Kersting, C. Plagemann, P. Pfaff, and W. Burgard. Most likely heteroscedastic gaussian process regression. In Proceedings of the 24th International Conference on Machine Learning, ICML ’07, pages 393–400, New York, NY, USA, 2007. ACM.

[2] M. Balandat, B. Karrer, D. R. Jiang, S. Daulton, B. Letham, A. G. Wilson, and E. Bakshy. BoTorch: A Framework for Efficient Monte-Carlo Bayesian Optimization. Advances in Neural Information Processing Systems 33, 2020.

[3] S. Cakmak, R. Astudillo, P. Frazier, and E. Zhou. Bayesian optimization of risk measures, 2020


**Time Spent Reviewing:**

2

---

> ### Author Response · Authors · 2021-08-10
> **Response to Reviewer hxC9**
>
> We thank the reviewer for the thorough and insightful feedback. We incorporate references and edits in the revised paper and also answer the most detailed questions below.
>
> Thanks for the references on CVaR and VaR, we agree that these are also important metrics for choosing risk-averse solutions. Which one is preferred depends on the application at hand, and there is no general consensus. Similarly, the application at hand defines the risk level parameter for CVaR or coefficient of absolute risk tolerance for RAHBO. Our mean-variance approach provides a principled method for a well-accepted risk metric and thus expands the toolbox of risk-averse Bayesian optimization. We find this line of research interesting to compare with, though one should overcome the following differences in the use cases of the existing literature [17, 9, 30] to set a fair comparison with RAHBO. Particularly, in these works, the environmental variable $w$ is assumed to have known distribution (might be restrictive), it is independent of $x$ (homoscedastic only) and controllable at the experimentation phase (may not be realistic when it comes to noise).
>
> We find interesting the reviewer's ideas about sampling efficiency as well the additional reference [1] where a single evaluation is used to model the noise variance. Our setup with $k$ evaluations clearly leads to the cost increase, which however allows for the better noise variance model, crucial for the risk-averse setting. This non-trivial trade-off for k is incurred in regret guarantees for RAHBO and such guarantees are not obvious for [1]. We agree that their elegant way of incorporating the smoothness assumption on the variance proxy is a promising direction for future work on sample efficiency.
>
> Finally, the modeling setup for RAHBO practically coincides with HeteroskedasticSingleTaskGP in Botorch and can be used as an alternative.

---

### Official Review · Reviewer_ZDq3 · 2021-07-16

**Rating:** 5
**Confidence:** 4

**Summary:**

This paper proposes a risk-averse Bayesian optimization algorithm, which aims to find an input with both large expected return and small noise-dependent noise variance. Both the mean function and input-dependent noise variance are assumed to be unknown functions from a reproducing kernel Hilbert space, and hence both need to be estimated online. Both theoretical guarantees and empirical evaluations are given.

**Limitations And Societal Impact:**

Yes.

**Main Review:**

This paper aims to tackle an important problem, i.e., designing BO algorithms that are risk-averse in terms of the risk introduced by input-dependent noise. The approaches taken by the paper, both theoretical and empirical, are all natural and reasonable. However, I have some concerns/questions regarding the theoretical and empirical results of this paper, please see the details below.

Weaknesses/questions:
- It looks like the Gaussian process posterior mean and variance for heteroscedastic GP adopted in this paper (equations 5 and 6) have a few differences between their homogenous counterpart (e.g., refer to equations 2 and 3 in reference [12]), even when a homogeneous noise variance is used, equations 5 and 6 in the current paper don't reduce to equations 3 and 3 in [12]. For example, the noise variance doesn't appear in the homogeneous case given in [12], but appears in the heteroscedastic case in equations 5 and 6 in the current paper. Is there an intuitive explanation for this difference?
- Proposition 1: I have a few questions regarding these theoretical results. Firstly, $\beta_T$ defined in equation 8 seems to be data-dependent, i.e., it depends on the history of selected input locations. So, I think a concrete expression regarding the growth rate of $\beta_T$ should be given, in order to support the claim that the proposed algorithms are no-regret (as claimed in line 67). Next, is there an intuitive interpretation regarding the dependence on $\lambda$? What is the recommended value of $\lambda$? In the homogeneous case like [12], $\lambda$ is set to a value dependent on $T$, but here it seems like a free parameter. In fact, the same questions apply to Theorem 2 as well.
- Lines 164-165: it says here that "the same upper bounds are still applicable in the considered heteroscedastic case (i.e., by setting $\sigma^2$ to $\varrho^2$)". If I understood correctly, if we set $\sigma^2$ to $\varrho^2$, the corresponding information gain (see the formula on line 160) will be smaller than the information gain with heteroscedastic noise variance, since the information gain is inversely related to the noise variance. That means the max information gain derived using $\varrho^2$ is smaller than by the $\gamma_T$ from Proposition 1, so that doesn't really give us a concrete regret upper bound in terms of homoscedastic noise variance. Does this contradict what is said here in these few lines, or is my understanding inaccurate?
- Equation 15: what's the intuition behind the use of $\text{ucb}^{var}_t(x_1)$ here? BTW I think the second $x_1$ should be $x_t$.
- Theorem 2: in addition to the questions I have regarding the dependence on $\beta_T$ and $\lambda$ as mentioned above, the dependence on $k$ here seems a little counter-intuitive: why is the regret bound increasing in $k$? Does that suggest we should set $k$ to the smaller value of 1?
- Corollary 2.1: does this result hold "with high probability"?
- Experiments: does the horizontal axis in Figure 2 (and other figures) consider the value of $k$? That is, if $k=10$ experiments are repeated for every evaluated input, does that correspond to 1 iteration or 10 iterations in the figure? If it corresponds to 1 iteration, the comparison may not be fair for GP-UCB. Moreover, I think an ablation w.r.t. the value of $k$ is needed, because I think it will have a huge impact on the estimated variance. For example, if $k$ is larger, I suppose RAHBO-US will perform better.
- lines 221-224: are the initial steps for variance estimation included in the comparisons shown in the figures?
- lines 250-251: why does standard GP-UCB prefer noisier inputs? Shouldn't it be indifferent between different inputs with the same expected function value?
- Figure 5: it seems that the advantage of the proposed method isn't so clear in these figures, especially in the hyperparameter tuning tasks of Figures b and c.
- Lines 268-269: are you trying to refer to Figure 4 instead of Figure 10 here? Because if it's Figure 10, the word "drastically" (line 271) will be an overstatement.

Some more minor points:
- Figure 2 caption, third and fourth lines: I think (c) and (d) are reversed.
- I think Figure 3 is never referred to in the text.

**Time Spent Reviewing:**

6 hours

---

> ### Author Response · Authors · 2021-08-10
> **Response to Reviewer ZDq3**
>
> We would like to thank the reviewer for constructive feedback.
>
> Firstly, we address the main questions regarding the theoretical part.
> - The different GP posterior updates result from the prior and noise models that are a design choice within the algorithm,  while the corresponding concentration bounds are the ones that matter the most for obtaining regret bounds. These modeling assumptions for RAHBO and [12] are different, e.g., the noise variance in the homogeneous case in [12] is explicitly incorporated in the prior.
> - Similarly, the strategy for setting $\lambda$ for RAHBO and [12] differs due to the modeling assumptions. An extra example of such a difference for $\lambda$ can be found in two papers with homoscedastic case [12] (set to $1+1/t$) and [Durand2018] (set to $\sigma^2$). In our case, it is indeed a positive constant that we set to $1$ in our experiments (similar to [19]). As to the explicit appearance of $\lambda$ in the regret, thanks to the reviewer’s feedback we discovered a typo that propagates through the analysis, influencing however only the constants but not the main scalings. By fixing this, the only dependence on $\lambda$ in the regret bound is left inside the $\beta_T$ parameter. We thank the reviewer for carefully going through the dependencies and constants.
>
> - Next, we clarify the bounds for the data-dependent $\beta_T$ and information gain  $\gamma_T$ that also appear in the regret bound. Firstly, $\beta_T$ can be easily bounded in terms of the information gain $\gamma_T$, and such scaling can be found, for example, in [12,19]. In short, it’s achieved by expanding the entropy definition through the matrices $K_T$ and $\Sigma_T$ rather than $\rho(x_t)$ and $\sigma_t(x_t)$.
> Secondly, the inaccuracy in the manipulation with $\gamma_T$ comes from the fact that the substitution happens not only in the denominator of the formula on line 160 but also in the numerator. Particularly, the posterior variance also depends on $\rho^2(x)$ in Eq.6. A careful manipulation with these terms leads to the correspondence between the heteroscedastic and homoscedastic information gains where for the latter the upper bounds are widely known. This correspondence works under mild assumptions for $\rho(x)$, namely its bounds $\min_{x_t}\rho(x_t)$ and $\max_{x_t}\rho(x_t)$, i.e., affects only the constants and does not change the scaling. We will make the derivation explicit in the revised version to avoid any further confusion.
> - Finally, the regret dependence on $k$ incorporates a non-trivial trade-off: Larger $k$ allows for better estimation of the noise model but it also increases sampling complexity, and, hence, setting $k=2$ does not necessarily optimize the regret bound. We leave the question of finding optimal $k$ for future work.
>
> - In Equation 15, where it should indeed be $ucb_t^{var}(x_t)$, $ucb_t^{var}(x)$ is used as the original matrix $\Sigma_T$ in (5)-(6) cannot be computed with the unknown variance-proxy. At the same time, for the confidence bounds in Lemma 1 to hold, it suffices to use an upper bound of the unknown variance-proxy.
>
> - The result of Corollary 2.1 holds with high probability and the proof can be found in Appendix A.6.
>
> We now clarify the questions regarding the experimental part:
>
> - One BO iteration in all figures corresponds to k evaluations. GP-UCB also uses a heteroscedastic noise model that relies on both sample mean and variance, i.e., exploiting these k evaluations. That makes the comparison across the methods fair as they use the same information in the best possible way. We agree with the reviewer about the ablation study as we see in the regret bounds the non-trivial trade-off for k. The same, however, holds for the RAHBO-US method, where the trade-off appears at the uncertainty sampling stage.
> - Lines 221-224: For the sake of the fair comparison, we indeed plot the initial steps for model initialization (same for all methods) as well as for variance estimation (for RAHBO-US). For example, in Figure 2, the RAHBO-US line starts later than others due to the additional budget for the uncertainty sampling.
> - Lines 250-251: We answer the similar question of Reviewer UhGC regarding Figure 2. Namely, the intuition for the behavior in Figure 2 relies on the two following facts: Querying noisier evaluations leads to wider confidence intervals in the heteroscedastic GP. GP-UCB does not distinguish explicitly between the regions with different noise levels. As a result, GP-UCB samples points in the noisy region to shrink the UCB while our method relies on a simple regularization against it.
> - We demonstrate the results on two synthetic functions and two real-world datasets. RAHBO indeed demonstrates more impressive results for the FEL application. That can be explained by two factors: noise in the physical systems might be more substantial and the results for RF are already decent.
> - The reviewer is correct that these statements are about Fig 4.
>
> [Durand2018] A. Durand, O.A. Maillard, J. Pineau. Streaming kernel regression with provably adaptive mean, variance, and regularization. The Journal of Machine Learning Research 19 (1). 2018

---

> > ### Comment · Reviewer_ZDq3 · 2021-08-20
> > **Further Clarifications**
> >
> > I appreciate the authors' response. However, there are still a few points where my concerns are not completely addressed and may require further clarifications.
> >
> > - Regarding the value of $\lambda$ (the second bullet point in your response above), it still seems strange to me that $\lambda$ is left as a free parameter in the final regret bound. As the authors have pointed out, it is usually set to some concrete values (rather than a free parameter that can be chosen by the user) which may differ due to different modelling assumptions, e.g., [12] sets it to $1+1/T$ and [Durand2018] sets it to $\sigma^2$. Could you give more explanations on this difference?
> >
> > - Regarding the value of $\beta_T$ (the third bullet point in your response above), the authors have mentioned that $\beta_T$ can be easily bounded in terms of $\gamma_T$ following [12,19]. Can you please provide the concrete upper bound on $\beta_T$ in terms of $\gamma_T$? Moreover, also related to this point, my previous question regarding how to "support the claim that the proposed algorithms are no-regret (as claimed in line 67)" was not addressed (see my second bullet point in the original review). I think in order to support the claim of no regret, a concrete (not-data-dependent) upper bound on $\beta_T$ is needed.
> >
> > - Also related to your third bullet point in your response above, if I understand correctly, the response says that if we set the noise variance $\sigma^2$ to $\varrho$, the resulting homogeneous max information gain is in fact an upper bound on the heterogeneous max information gain? Again I think a detailed derivation is needed, because I think otherwise, we cannot claim that the algorithm is no-regret (because to the best of my knowledge, the growth rate is known only for homogeneous max information gain, not for the heterogeneous one).
> >
> > - The other points in the response all seem fair to me, although for the experiments, I think only showing that the proposed algorithm works competitively in one real-world experiment (the FEL experiment) may not be enough. Also, I think a discussion on the trade-off regarding the value of $k$ should be added to the paper.

---

> > > ### Author Response · Authors · 2021-08-25
> > > **Response for the Further Clarifications**
> > >
> > > We appreciate the reviewer's response. Below, we provide more details on each of the questions.
> > >
> > > - (On $\lambda$) Since the regret bounds are not optimized in terms of $\lambda$ in the homoscedastic setting (note that both $\gamma_T$ and $\beta_T$ depend on $\lambda$), different works set $\lambda$ to different constant values.
> > > Besides the previous two values, e.g., [6] use $\lambda=1$, and [*] use $\lambda \geq 0$. In turn, setting $\lambda$ to a constant value (independent of $T$) does not affect the main scalings, as this only impacts constants in regret bounds. As mentioned in the previous response, we will explicitly set $\lambda =1$ in our theoretical results, as we also use this value in our experiments.
> > >
> > > [*] Sayak Ray Chowdhury, Aditya Gopalan. On Batch Bayesian Optimization. 2019
> > >
> > >
> > > - (On $\beta_T$) Following our modelling assumptions $f_{1:T} \sim \mathcal N (0, \lambda^{-1}K_T)$ and $\xi_{1:T} \sim \mathcal N (0, \Sigma_T)$, the information gain $I(y_{1:T}, f_{1:T}) = H(y_{1:T}) - H(y_{1:T}|f_{1:T}) $ is given as follows:
> > > $$I(y_{1:T}, f_{1:T})  = \underbrace{\frac{1}{2} \ln \big(\det (2\pi e (\lambda^{-1}K_T + \Sigma_T) ) \big)}\_{H(y_{1:T})} - \underbrace{\frac{1}{2} \ln \big(\det (2\pi e \Sigma_T) \big)}\_{H(y_{1:T}|f_{1:T})} = \frac{1}{2} \ln \frac{\det(K_T +  \lambda\Sigma_T)}{\det (\lambda\Sigma_T)}.$$
> > >
> > > By definition then $\gamma_T = \max_{A_T} I(y_{1:T}, f_{1:T}) \geq \frac{1}{2} \ln \frac{\det(K_T +  \lambda\Sigma_T)}{\det (\lambda\Sigma_T)}$.
> > > On the other hand, $\beta_T$ defined in Lemma 1 can be expanded in a data-independent manner as follows:
> > >
> > > $$\beta_T := \sqrt{2\ln \bigg( \frac{\det(\lambda \Sigma_T +K_T)^{1/2}}{\delta\det(\lambda \Sigma_T)^{1/2}}}
> > >         \bigg) + \sqrt{\lambda}\|f\|_{\kappa}  =  \sqrt{2\ln \frac{1}{\delta} + \ln \frac{\det(\lambda \Sigma_T +K_T)}{\det(\lambda \Sigma_T)}} +  \sqrt{\lambda} |f|_k \leq \sqrt{2\ln \frac{1}{\delta} + \gamma_T} + \sqrt{\lambda}B_f.$$
> > >
> > > - (On no-regret) Above we show that $\beta_T$ can be bounded in a data-independent manner, as long as upper bounds on $\gamma_T$ are readily available. The latter follows from bounding $\gamma_T$ in the heteroscedastic setting in terms of known bounds [35] for the homoscedastic case, as we discuss below.
> > >
> > >
> > > - (On $\gamma_T$) Here we provide the detailed derivation. For simplicity, let us introduce the notation $\gamma_T^{\rho_x}$ for the maximum information gain for heteroscedastic noise with variance-proxy $\rho^2(x)$. Similarly, $\gamma_T^{\sigma}$ denotes the maximum information gain for homoscedastic noise with fixed variance-proxy $\sigma^2$. Assume that $\varrho^2(\cdot) \in [\smash{\underline{\varrho}^2}, \bar{\varrho}^2]$ for some constant values $\bar{\varrho}^2 \geq \smash{\underline{\varrho}^2} > 0$.
> > >
> > > Below, we show that $\gamma_T^{\rho_x} \leq  \gamma_T^{\sigma}  \frac{\bar{\varrho}^2}{\smash{\underline{\varrho}^2}} $ when $\sigma^2:= \bar{\varrho}^2 $  which thus only affects the constants but not the main scaling (in terms of $T)$ of the known bound for the homoscedastic maximum information gain.
> > >
> > > $$\gamma_T^{\rho_x}
> > >     \underbrace{=}\_{(1)}
> > >         \max_{A_T} \frac{1}{2}\sum_{t=1}^T \ln \bigg( 1+ \frac{\sigma^2_{t-1}(x_t | \rho^2(x_t))}{ \rho^2(x_t)} \bigg)
> > >     \underbrace{\leq}\_{(2)}
> > >         \max_{A_T} \frac{1}{2}\sum_{t=1}^T \ln \bigg( 1+ \frac{\sigma^2_{t-1}(x_t | \bar{\varrho}^2)}{ \smash{\underline{\varrho}^2}} \bigg)
> > >     \underbrace{=}\_{(3)}
> > >         \max_{A_T} \frac{1}{2}\sum_{t=1}^T \ln \bigg( 1+ \frac{\bar{\varrho}^2}{\smash{\underline{\varrho}^2}} \frac{\sigma^2_{t-1}(x_t | \bar{\varrho}^2)}{\bar{\varrho}^2} \bigg) $$
> > > $$  \underbrace{\leq}\_{(4)}
> > >         \max_{A_T} \frac{1}{2}\sum_{t=1}^T \frac{\bar{\varrho}^2}{\smash{\underline{\varrho}^2}} \ln \bigg( 1+ \frac{\sigma^2_{t-1}(x_t | \bar{\varrho}^2)}{\bar{\varrho}^2} \bigg)
> > >     \underbrace{=}\_{(5)}
> > >         \max_{A_T} \frac{\bar{\varrho}^2}{\smash{\underline{\varrho}^2}} \frac{1}{2}\sum_{t=1}^T \ln \bigg( 1+ \frac{\sigma^2_{t-1}(x_t | \sigma^2)}{\sigma^2} \bigg)=  \frac{\bar{\varrho}^2}{\smash{\underline{\varrho}^2}} \gamma_T^{\sigma}, $$
> > >
> > > where in (1) we highlight the dependence of $\sigma^2_t(\cdot|\rho^2(x_t))$ on $\rho^2(x_t)$ and define information gain $I(y_{1:T}, f_{1:T})$ with $y_t|y_{1:t-1} \sim \mathcal{N} (\mu_{t-1}(x_t), \rho^2(x_t) + \sigma^2_{t-1}(x_t))$ as follows:
> > >
> > > $$I(y_{1:T}, f_{1:T}) = \underbrace{\frac{1}{2}  \sum_{t=1}^T\ln \bigg(2\pi e   \rho^2(x_t) \bigg)  + \frac{1}{2}\sum_{t=1}^T \ln \bigg( 1+ \frac{\sigma^2_{t-1}(x_t)}{\rho^2(x_t)} \bigg)}\_{H(y_{1:T})}- \underbrace{\frac{1}{2} \sum_{t=1}^T \ln (2\pi e   \rho^2(x_t))}\_{H(y_{1:T}|f_{1:T})} = \frac{1}{2}\sum_{t=1}^T \ln \bigg( 1+ \frac{\sigma^2_{t-1}(x_t )}{ \rho^2(x_t)} \bigg). $$
> > >
> > > In (2), we lower bound the denominator $\rho^2(x_t)$ and upper bound the numerator $\sigma^2_{t-1}(x_t | \rho^2(x_t))$ (due to monotonicity w.r.t. noise variance., i.e., $\sigma^2_{t-1}(x_t| \Sigma_t) \leq \sigma^2_{t-1}(x_t| \bar{\varrho}^2 \mathbf I_t)$). In (3) we multiply by $1=\bar{\varrho}^2/\bar{\varrho}^2$. In (4) we use Bernoulli inequality since $\bar{\varrho}^2/\smash{\underline{\varrho}^2} \geq 1$. The obtained expression can be interpreted as a standard information gain for homoscedastic noise and, particularly, in (5) the variance-proxy $\sigma^2:= \bar{\varrho}^2 $, thus completing the proof. Finally, we will include and elaborate on this in the final paper version.
> > >
> > > - We thank the reviewer for the useful suggestions. We will incorporate such a discussion in the main body of our paper.

---

> > > > ### Comment · Reviewer_ZDq3 · 2021-09-02
> > > > **Further Response**
> > > >
> > > > I appreciate the authors' efforts in deriving the additional theoretical results. As far as I can tell, the results in the response above are all correct and have resolved all my previous concerns on $\beta_T$ and the maximum information gain.
> > > >
> > > > As a few suggestions, because the growth rates are only known for max information gain with homoscedastic noise, I would suggest the authors to revise the main paper to present the regret bounds (Proposition 1 and Theorem 2) in terms of $\gamma_T^{\sigma}$ (following the notation in the response). I would also suggest replacing $\beta_T$ in these regret bounds by its upper bound as the authors have derived in the response above, because having $\beta_T$ in the regret bound as in the current version of the paper doesn't give readers an intuitive understanding of the concrete dependence it brings. After these changes, I think the theoretical results would be much more complete, and then the authors could discuss in the paper when the algorithm is no-regret by making use of existing results on the growth rate of $\gamma_T^{\sigma}$.
> > > >
> > > > As an additional comment, if I understand correctly, the regret upper bound now depends linearly on the term $\frac{\overline{\varrho}}{\underline{\varrho}}$ (because $\beta_T$ would bring an additional dependence on $\sqrt{\gamma_T}$ in addition to the existing $\sqrt{\gamma_T}$ term). I believe this is something new that's not already in the paper, especially the inverse dependence on $\underline{\varrho}$. This new dependence suggests that the smaller the minimum variance proxy, the worse the regret. This is a little counter-intuitive and hence should be discussed in the paper.
> > > >
> > > > In light of these, I've increased my score to 5. I'm hesitant to increase further mainly due to my earlier concern on empirical results (last point under "Further Clarifications" above), but I don't have any other major concerns.

---

> > > > > ### Author Response · Authors · 2021-09-02
> > > > > **Final Response to Reviewer ZDq3**
> > > > >
> > > > > We would like to thank the reviewer for the comments and suggestions.
> > > > > We will improve the exposition accordingly.

---

### Official Review · Reviewer_UhGC · 2021-07-16

**Rating:** 6
**Confidence:** 4

**Summary:**

The paper proposes a Bayesian optimization method for robust optimization of black-box functions under heteroskedastic observation noise. The method uses an objective that trades off mean and noise variance to identify solutions with high mean and low noise variance. The authors propose modeling the black-box function and the black-box noise variance function using independent GPs and using a UCB-style acquisition function. The authors prove a sublinear cumulative regret bound for the method.

**Limitations And Societal Impact:**

Limitations are discussed and potential future directions are interesting. “We are not aware of any societal impacts of our work” – this (as with an optimization algorithm) could be used for nefarious endeavors and could be discussed.

**Main Review:**

Strengths:
-	Theoretical grounding: the sublinear cumulative regret bound is nice
-	Under-explored and interesting problem class of robustness with respect to stochastic evaluations
  -	Many works on robust optimization consider robustness with respect to input noise rather than output noise
-	The FEL experiment is a nice demonstration

Weaknesses:
-	Typically, expected performance under observation noise is used for evaluation because the decision-maker is interested in the true objective function and the noise is assumed to be noise (misleading, not representative). In the formulation in this paper, the decision maker does care about the noise; rather the objective function of interest is the stochastic noisy function. It would be good to make this distinction clearer upfront.
-	The RF experiment is not super compelling. It is not nearly as interesting as the FEL problem, and the risk aversion does not make a significant difference in average performance. Overall the empirical evaluation is fairly limited.
-	It is unclear why the mean-variance model is the best metric to use for evaluating performance
  -	Why not also evaluate performance in terms of the VaR or CVaR?
  -	The MV objective is nice for the proposed UCB-style algorithm and theoretical work, but for evaluation VaR and CVaR also are important considerations

Writing:
-	Very high quality and easy to follow writing
-	Grammar:
  -	L164: “that that”
  -	Figure 5 caption: “Simple regret fat the reprted”

Questions:
-	Figure 2: “RAHBO not only leads to strong results in terms of MV, but also in terms of mean objective”? Why is it better than GP-UCB on this metric? Is this an artifact of the specific toy problem?


**Time Spent Reviewing:**

2

---

> ### Author Response · Authors · 2021-08-10
> **Response to Reviewer UhGC**
>
> We would like to thank the reviewer for the detailed comments.
>
> We indeed focus on settings where we care about the stochastic outcomes rather than the (deterministic) expected objective. While often the stochastic outcome is considered "noise", in many applications of interest (such as the ones considered in the paper), we care about the actual realized outcomes as opposed to the expected value.
>
> We agree with the reviewer that CVaR and VaR are also important metrics for choosing risk-averse solutions. Which one is preferred depends on the application at hand, and there is no general consensus.  Our approach expands the toolbox of risk-averse Bayesian optimization by providing a principled method for a well-accepted risk metric, the mean-variance tradeoff.
>
> The intuition for the behavior in Figure 2 relies on the two following facts: Querying noisier evaluations leads to wider confidence intervals in the heteroscedastic GP.  GP-UCB does not distinguish explicitly between the regions with different noise levels. As a result, GP-UCB samples points in the noisy region to shrink the UCB while our method relies on a simple regularization against it.

---

> > ### Comment · Reviewer_UhGC · 2021-08-10
> > **Response to authors**
> >
> > Thanks for the response. The explanation of Figure 2 makes sense.

---

### Official Review · Reviewer_2VoZ · 2021-07-20

**Rating:** 7
**Confidence:** 3

**Summary:**

This paper proposes a new UCB strategy for Bayesian optimization that focuses on mean-variance optimization instead of expected value. The authors argue that this setting is relevant for heteroscedastic settings where one seeks high mean values with lower risk. Regret bounds are given to provide theoretical results about the convergence of the proposed algorithm for this setting. Empirical results on two synthetic problems and two "real-world" problems show that the proposed approach returns solutions with a higher mean and lower risk than standard risk-neutral GP-UCB.

**Main Review:**

**Strengths**

A risk-averse strategy for Bayesian optimization is surely relevant in many applications. Noisy observations are ubiquitous in science and engineer problems. In fact, it is somehow surprising that previous work hasn't addressed this setting before. The "Tuning a Free Electron Laser" example was the most interesting experiment. I found this work fairly complete and relatively easy to understand, despite the heavy notation for the theorems.

**Weakness**

No major issues. I would only suggest that the empirical section could be more substantial if all the results were motivated by real applications or if the synthetic experiments were conducted over multiple test functions (and not only two). It would be interesting to see aggregate results across a more extensive set of problems.

**Minors**

- lines 41 (suggestion). Exploration vs. Exploitation for optimizing the expected performance; instead of an 'and optimizing.'
- Maybe I've missed that, but I was expecting RAHBO-US to execute x% observations of US and (1-x)% of RAHBO. What values of x were considered?

**After author's response**

Thanks for all the detail answers!

**Time Spent Reviewing:**

2

---

> ### Author Response · Authors · 2021-08-10
> **Response to Reviewer 2VoZ**
>
> We would like to thank the reviewer for the positive feedback and suggestions. We set the observation budget differently depending on the application (around 10% of the total BO iterations number). The specific values can be observed in the plots.

---

### Decision · Program_Chairs · 2021-09-27

**Decision:**

Accept (Poster)

**Comment:**

This paper makes an important contribution to the theory of Bayesian optimization. Namely, it provides a sublinear cumulative regret bound in the heteroscedastic setting, the first of its kind, and show that this regret bound additionally applies in the risk averse setting. This result is particularly significant given that the overwhelming majority of Bayesian optimization papers assume homoscedastic noise, which may arguably less common than the heteroscedastic setting. Be sure to incorporate suggested minor corrections by the reviewers in the final camera ready version.